# Applying Non-Parametric Bayesian Network to estimate monthly maximum river discharge: potential and challenges

Elisa Ragno[1], Markus Hrachowitz[1], and Oswaldo Morales-Nápoles[1]

[1]Delft University of Technology, Faculty of Civil Engineering and Geosciences, 2628 CN, Delft, Netherlands

**Correspondence:** Elisa Ragno (e.ragno@tudelft.nl)

**Abstract.** Non-Parametric Bayesian Networks (NPBNs) are graphical tools for statistical inference widely used for reliability analysis and risk assessment. Even with their several advantages, such as the embedded uncertainty quantification and limited computational time required for the inference process, the implementation of NPBNs in hydrological studies is still scarce. Hence, to increase our understanding of the applicability of NPBNs and extend their use in hydrology, in this study, we explore the potential of NPBNs to reproduce catchment-scale hydrological dynamics. Long-term data from 240 river catchments with contrasting climates across the United States from the Catchment Attributes and Meteorology for Large-sample Studies (CAMELS) data set will be used as actual means to test the utility of NPBNs as descriptive models and to evaluate them as predictive models for monthly maximum river discharge. First, we analyse the performance of two networks, one unsaturated (hereafter UN-1) and one saturated (hereafter SN-1), defined by hydro-meteorological variables and their bi-variate correlations. These networks, trained on the individual catchment, are used to generate monthly maximum river discharge considering the catchment as a single element. Then, we analyse the performance of a saturated network (hereafter SN-C), consisting of the network SN-1 and including physical catchments attributes, to model a group of catchments and infer monthly maximum river discharge in ungauged basins based on the similarity of the attributes. The results indicate that the network UN-1 is suitable for catchments with a positive dependence between precipitation and river discharge, while the network SN-1 can reproduce discharge also in catchments with negative dependence. Furthermore, in $\sim 40\%$ of the catchments analysed, the network SN-1 can reproduce statistical characteristics of discharge, tested via the Kolmogorov-Smirnov (KS) statistic, and have a Nash-Sutcliffe Efficiencies (NSE) $\geq 0.5$. Such catchments receive precipitation mainly in winter and are located in energy-limited regions at low to moderate elevation. Further, the network SN-C, in which the inference process benefits from information from similar catchments, can reproduce river discharge statistics in $\sim 10\%$ of the catchments analysed. However, in these catchments, a common dominant physical attribute was not identified. In this study, we show that once a NPBN is defined, it is straightforward to infer discharge when the remaining variables are known. We also show that it is possible to extend the network itself with additional variables, i.e. going from the network SN-1 to the network SN-C. However, the results also suggest considerable challenges in defining a suitable NPBN, particularly for predictions in ungauged basins. These are mainly due to the discrepancies in the time scale of the different physical processes generating discharge, the presence of a "memory" in the system, and the Gaussian-copula assumption used by NPBNs for modelling multivariate dependence.

## 1 Introduction

Strategies for water resources management and planning mostly rely on predictions from hydrological models (Hrachowitz and Clark, 2017). Such models are mathematical representations of the relationship between catchment structure and response be-
30 havior (Wagener et al., 2007). In the history of hydrological modelling, two main model philosophies can be identified: models aiming at explicitly representing physical processes at different degrees of complexity, hereafter referred to as process-based models, and process-agnostic models relying on relationships between one or multiple system input and output variables, e.g., precipitation and streamflow, without further assumptions on underlying mechanistic processes, hereafter data-driven models (Todini, 2011). A trade-off between what we defined process- and data-driven models is represented by the Data-Based
Mechanistic approach (DPM; Young and Beven, 1994) for modelling complex systems in hydrology, and in general. Such an approach looks for parametrically efficient, low order, dominant mode models identified and validated based on stochastic methods and associated statistical analysis (Young and Beven, 1994).

Data-driven models in general differ on the input-output technique implemented, which might not have a conventional physical interpretation (Todini, 2011), such as multilinear regression functions, (e.g., Barbarossa et al., 2017), artificial neural network,
(e.g., Beck et al., 2015), long short-term memory networks (e.g., Kratzert et al., 2019), and probabilistic graphical models (e.g., Paprotny and Morales-Nápoles, 2017). For river discharge prediction at longer time resolutions, such as monthly, data-driven models are the predominant models found in the literature (e.g., Barbarossa et al., 2017; Sivakumar et al., 2001; Ren et al., 2020; Fathian et al., 2019; Anmala et al., 2000; Wei et al., 2012; Fathian et al., 2019).

A wide range of scientific publications illustrates progress in formulations and implementations of both process-based and
45 data-driven hydrological models, highlighting their respective potentials. However, among data-driven models, less attention has so far been given to explicitly representing the interdependence between inflow and outflow via high dimensional probability functions. Bi- and multi-variate probability function, such as copulas, have been mostly implemented to derive critical flood design values when multiple flood characteristics are of interest (e.g., Salvadori and De Michele, 2004; Grimaldi and Serinaldi, 2006), or when flood events result from the interaction between multiple physical drivers (e.g., Moftakhari et al.,
2017; Bevacqua et al., 2017). Recently, vine-copula-based models for high dimensional probability, such as Nonparametric Bayesian Networks (NPBNs), have gained popularity in hydrological studies (e.g., Sebastian et al., 2017; Couasnon et al., 2018; Paprotny and Morales-Nápoles, 2017). Different applications of NPBNs can be found in the scientific literature, (e.g. Morales-Nápoles et al., 2014a; Jesionek and Cooke, 2007; Hanea and Ale, 2009; Kosgodagan-Dalla Torre et al., 2017). In reliability studies, Morales-Nápoles and Steenbergen (2014) implemented NPBNs for modelling complex traffic systems and
showed that they can be used for computing design values for individual axles, vehicle weight, and maximum bending moments of bridges within certain time intervals. In hydrological studies, Sebastian et al. (2017) adopted NPBNs for generating synthetic storm events along Galveston Bay (Texas) based on different tropical cyclone characteristics at landfall and demonstrated their ability to generate plausible boundary conditions for coastal riverine models for flood analyses. Similarly, Couasnon et al.

(2018) applied NPBNs to model and assess the impact of flooding generated by the interaction between coastal and riverine
drivers while accounting for the spatial dependence between river tributaries. Paprotny and Morales-Nápoles (2017) introduced
the use of NPBNs for river discharge mean annual maximum and return period estimation and showed results comparable to
physically-based models. NPBNs are probabilistic graphical models representing high dimensional probability distribution
functions of system properties with complex dependence structures (Hanea et al., 2015) and support probabilistic inference of
system characteristic(s) by conditioning on known characteristics (Kurowicka and Cooke, 2002).The joint probability distribu-
tion is determined by defining the dependence between pairs of variables. Such a non-parametric joint probability distribution
is then more flexible compared to a theoretical parametric multivariate distribution because the dependence between variables
is not fixed by the theoretical parametric model, but it depends on how the variables (nodes of the network) are connected
to each other (arcs and parenting order). NPBNs' potential resides in several characteristics: (i) the uncertainty quantification
is embedded in the model given that all the variables included in the network and contributing to discharge generation are
treated as random variables; (ii) all the variables, not only river discharge, can be inferred by conditioning on the remaining
variables; (iii) causal relationships between variables from prior knowledge can be imposed in the network but, at the same
time, unknown relationship can be learned; (iv) information from different catchments can contribute to improve inference; (v)
and the computational time is limited.

Starting from these premises, the main objective of this study is to further explore and test the suitability of NPBNs as a tool to
reproduce catchment-scale hydrological dynamics and to explore challenges involved when inferring monthly maximum river
discharge. More specifically, long-term data from 240 river catchments across the United States from the Catchment Attributes
and Meteorology for Large-sample Studies (CAMELS, Newman et al., 2015; Addor et al., 2017) data set will be used as actual
means to test the utility of NPBNs as descriptive models and to evaluate them as predictive models for monthly maximum river
discharge considering the catchments individually and in group, to explore catchment similarity.

## 2   Catchments and data

For this study, we make use of the CAMELS data set (Newman et al., 2015; Addor et al., 2017). CAMELS provides homog-
enized long-term hydro-meteorological data and catchment attributes of catchments across the contiguous United States. To
limit potentially adverse effects of spatial heterogeneity, we analyse 240 catchments from the CAMELS data set with areas
$\leq 200\,\mathrm{km}^2$ (see Supplementary Material Table S1). For each selected catchment, we considered hydro-meteorological data and
catchment attributes in Table 1. As the objective of this study is to model maximum monthly discharge from 1980 to 2013,
we further process daily hydro-meteorological data as follows: (1) extract monthly maximum discharge from daily specific
discharge; (2) extract maximum daily precipitation over the previous 7 days from the day of the occurrence of the maximum
discharge, and (3) calculate the mean over the previous 7 days from the day of the occurrence of the maximum discharge value
of the remaining daily variables. Consequently, we generate a multidimensional data set in which all the variables are related
to the occurrence of the discharge event. The selection of this concomitant variables came after a preliminary investigation
of the strength of the correlation between maximum discharge and both maximum and cumulative precipitation over different

time windows, Supplementary Material Fig. S1. In addition, we investigate whether the maximum precipitation event extracted over the 7 days prior to the monthly maximum discharge is also the maximum precipitation event occurring that month. We observe that this is the case almost every months for stations at low to moderate altitude (Supplementary Material, Fig. S2),

supporting the assumption that in such catchments monthly maximum discharge is mainly driven by monthly maximum precipitation event. Such data pre-processing aims to generate a multivariate time series with independent and identically distributed (iid) observations. By selecting monthly maximum discharge, we assume that such discharge peaks, and corresponding hydro-meteorological variables, result from different underlying weather events. However, in particular discharge data do, inevitably and as a result of catchment memory effects, show some degree of autocorrelation (Supplementary Material, Fig. S3), which

might affect the correlation strength with the remaining variables. We will further discuss this aspect in the discussion section. Catchments attributes from the CAMELS database were used without further processing. The attribute aridity, $Ar$[-], refers to the ratio of long-term means of potential evapotranspiration calculated using Priestley-Taylor formulation and precipitation, where values higher/lower than 1 indicate water/energy-limited regions. The attribute precipitation seasonality $p_s$[-] (Woods, 2009) describes the temporal concentration of intra-annual precipitation occurrence and takes positive/negative values when

precipitation peaks occur in summer/winter. For further details on catchment attributes and their derivation, the reader is referred to CAMELS database documentation Addor et al. (2017).

**Table 1.** hydro-meteorological data and catchment attributes used in this study

| Data type | Unit | Symbol | Estimated Monthly Value | Original Resolution |
|---|---|---|---|---|
| Specific Discharge | $mm/day$ | $Q_{max}$ | daily max | daily |
| Temperature | $°C$ | $T$ | mean over 7 days prior $Q_{max}$ | daily |
| Precipitation | $mm/day$ | $P_{max}$ | max over 7 days prior $Q_{max}$ | daily |
| Shortwave downward radiation | $W/m^2$ | $R$ | mean over 7 days prior $Q_{max}$ | daily |
| Water vapor pressure | $Pa$ | $V_p$ | mean over 7 days prior $Q_{max}$ | daily |
| Monthly Runoff coefficient | $-$ | $C_m$ | ratio monthly discharge and cumulative precipitation | daily precip. and discharge |
| Elevation | $m.a.s.l$ | $Elv$ | $-$ | constant |
| Slope | $m/km$ | $Slp$ | $-$ | constant |
| Aridity | $-$ | $Ar$ | $-$ | constant |
| Precipitation seasonality | $-$ | $p_s$ | $-$ | constant |
| Fraction of forest | $-$ | $ff$ | $-$ | constant |

Catchments located in Eastern and Central-Eastern U.S. (56 %) are characterized by an average size of about 94 $km^2$ and average daily specific discharge of 1.3 $mm/day$ (Fig. 1a). These catchments are mostly situated at moderate elevations (average altitude 304 m.a.s.l.) and in energy limited areas ($Ar \sim 0.77$, Fig. 1d-b) with little precipitation seasonality ($p_s \sim 0.09$,

Fig. 1c). In contrast, catchments located in Western and Central-Western U.S. (44 %) have an average size of about 61 $km^2$

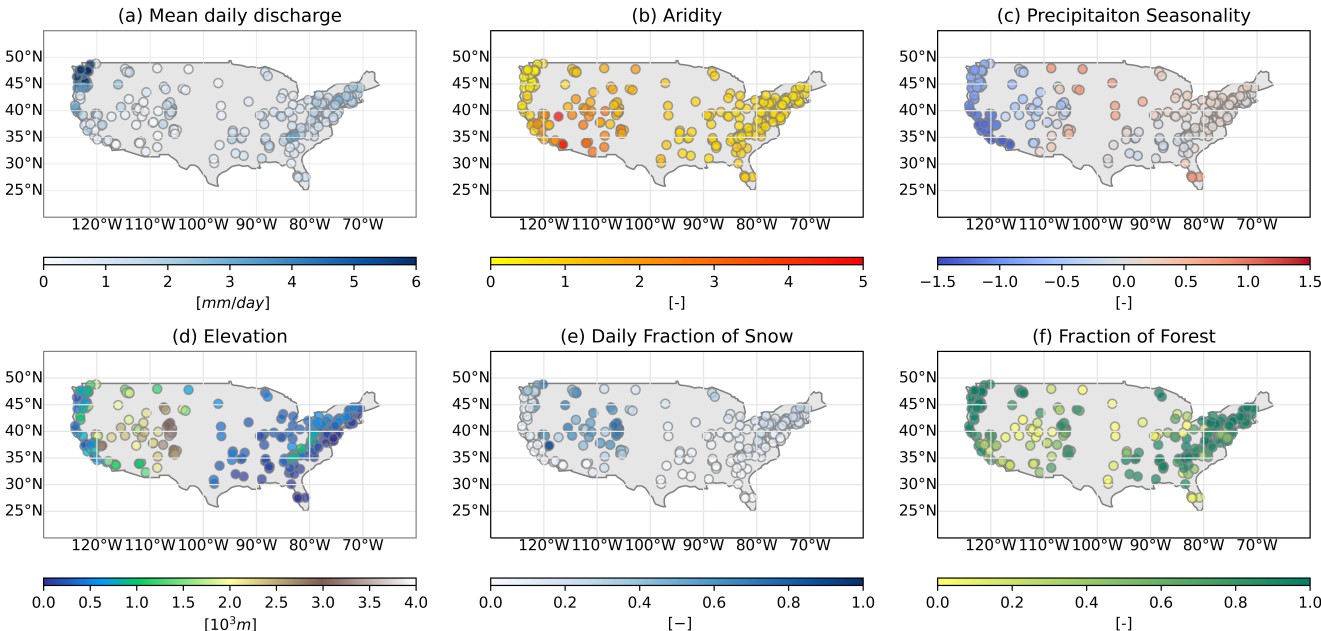

**Figure 1.** Catchment attributes extracted from the CAMELS database: (a) Mean daily discharge [mm/day]; (b) aridity as PET/P; (c) precipitation seasonality where positive (negative) indicates precipitation peaks in summer (winter); (d) elevation [m a.s.l.]; (e) daily fraction of snow indicates the fraction of precipitation falling as snow in case of temperatures below zero; (f) fraction of forest.

and are on average located at higher elevations with $\sim$760 m.a.s.l. in Western U.S and 2300 m.a.s.l. in Central-Western region, where precipitation falls mostly over winter ($p_s$ between $\sim$-0.9 and -0.2), Fig. 1d. While catchments in Western U.S. are on average located in energy-limited areas (Ar $\sim$ 0.6), they are located in water-limited regions in Central-Western U.S. (Ar $\sim$ 1.7, Fig. 1b). This difference is reflected in the mean daily discharge which is 3.1 and 0.64 $mm/day$ respectively, Fig. 1a.

Catchments in Central-Western U.S., given their elevation, have the highest ratio of daily precipitation falling as snow in a day with temperatures below zero ($\sim$ 0.5; Fig. 1e - daily fraction of snow).

   In the majority of the catchments selected (86 %), the correlation between monthly maximum discharge ($Q_{max}$) and maximum precipitation over 7 days ($P_{max}$) is positive, meaning that discharge is mainly driven by precipitation runoff (Fig. 2a). Catchments with negative correlation are mostly located in water-limited regions and at elevations above 1500 m.a.s.l., Fig.

2b-a. Furthermore, in such catchments, precipitation occurs mainly in winter and fraction of snow is greater than 0.4 (Fig. 2e-f).

   Hydro-meteorological variables and catchments attributes described so far are used in the following as input to reproduce catchment-scale hydrological dynamics via NPBNs.

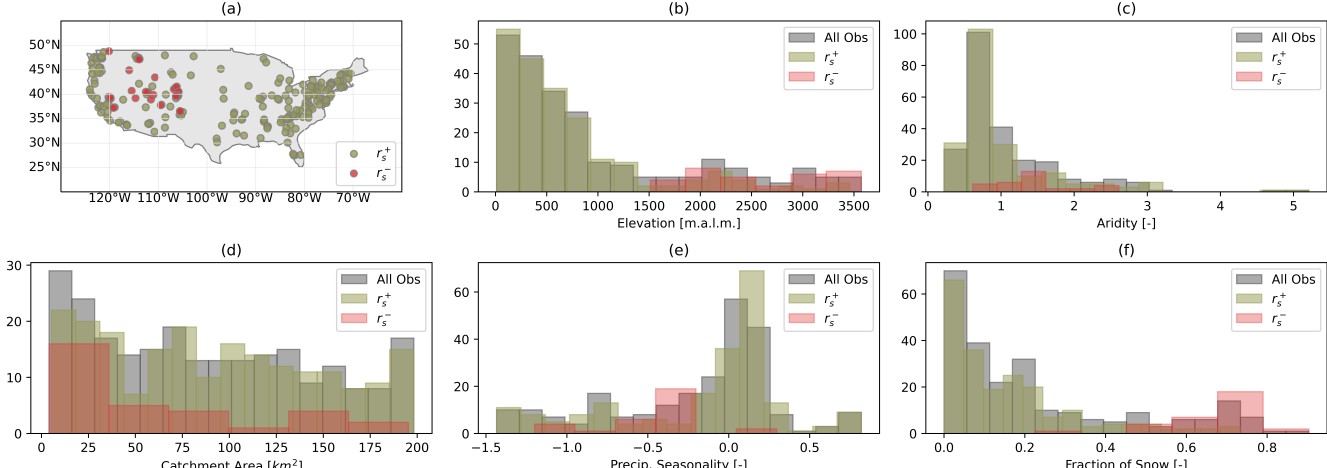

**Figure 2.** Catchment attributes based on the correlation between monthly maximum discharge, $Q_{max}$, and maximum precipitation over the previous 7 days, $P_{max}$. Panel (a) shows the geographical location of catchments with negative correlation between $Q_{max}$ and $P_{max}$, red dots, and catchments with positive correlation, green dots. Panel (b) shows the distribution of the attribute elevation of catchments with negative (red) and positive (green) correlation against the overall distribution (grey). Panels (c) to (f) show the same comparison as panel (b) but of the following attributes aridity, area, precipitation seasonality, and fraction of snow respectively.

## 3 Probabilistic Graphical Models: Bayesian Networks

Pearl (1985) first formalized the term Bayesian Network (BN) as a class of networks represented by influence diagrams or networks to model the probabilistic relationship between variables. Afterwards, BNs became a popular tool for dealing with uncertain domains (Aguilera et al., 2011).

A BN is defined by two components (Aguilera et al., 2011): a qualitative component, being a Directed Acyclic Graph (DAG) where the nodes are the random variables of the model and the arcs connecting two nodes indicate their statistical dependence;

and a quantitative component being the conditional distribution of each variable (child) given its direct preceding variables (parents). Given a network of $n$ nodes (variables) $\{X_1, \cdots, X_n\}$ and a set of parent nodes $S_i$ for node $i$, then the joint density (mass in the discrete case) is defined as:

$$f(x_1, ..., x_n) = \prod_{i=1}^{n} f_{x_i|S_i}(x_i|S_i) \tag{1}$$

The (conditional) independence relationships embedded in the probabilistic model can be easily visualized in the graphical

representation of the network (Pearl, 1985). Moreover, the absence of an arc guaranties the conditional independence between two "source" variables (Hanea et al., 2015), while the direction of the arc indicates the "flow of information" (Vogel et al., 2014). Strictly speaking, probabilistic dependence does not have a "direction". However, when it can be easily related to causality it is convenient to think of a "flow of information".

BNs differ on how nodes and arcs are quantified, and the inference process depends on this quantification. Discrete BNs specify

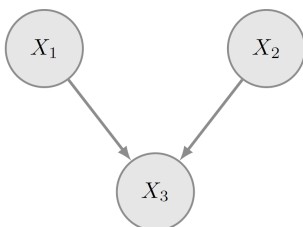

**Figure 3.** Illustrative Bayesian Network with 3 nodes

the source nodes, i.e., nodes without parents, as discrete random variables and conditional probability tables for child nodes (Hanea et al., 2006). Hybrid BNs (HBNs) involve both discrete and continuous variables. HBNs specify marginal distributions for nodes without parents and conditional distributions for child nodes. HBNs can be fully parametric, in which marginals and joint probabilities are from parametric families, or fully discrete, in which continuous variables are discretized (Hanea et al., 2015). Discretization of continuous variables, however, has the drawback of requiring a very large number of partitions to

guarantee a good approximation of the variables.

Figure 3 is an illustrative example of the qualitative component of a BN for the variables $\{X_1, X_2, X_3\}$. The quantitative component is given by $f_{X_1}(x_1)$, $f_{X_2}(x_2)$, and $f_{X_3|S_3}(x_3|S_3)$ where $S_3 = \{X_1, X_2\}$ is a set containing the parents of node $X_3$. The joint probability resulting from the above information is $f(x_1, x_2, x_3) = f_{X_1}(x_1) \cdot f_{X_2}(x_2) \cdot f_{X_3|S_3}(x_3|S_3)$. From Figure 3 it is possible to determine the dependence relationships between the nodes. The absence of the arc connecting $X_1$ and $X_2$

implies their independence ($X_1 \perp X_2$). However, $X_1$ and $X_2$ are conditional dependent when information on $X_3$ becomes available, i.e., $X_1$ and $X_2$ are dependent given $X_3$ ($X_1 \not\perp X_2|X_3$).

In the scientific literature, BNs have been implemented in multiple fields. Weber et al. (2012) reviewed BNs applications in reliability, risk, and maintenance areas and showed that BNs are tools able to address industrial system modelling in relation to increase complexity. Aguilera et al. (2011) reviewed the implementation of BNs in environmental sciences and concluded that

their application is still scarce due to the necessity of discretizing continuous variables and the limited availability of software. In a more recent application of BNs on natural hazards estimation, Vogel et al. (2014) showed their flexibility and applicability through three real case studies highlighting their ability to express information flow and independence assumptions between candidate predictors.

### 3.1   Nonparamentric Bayesian Networks

Kurowicka and Cooke (2005) introduced a vine-copula based approach for HBNs called Nonparametric Bayesian Networks (NPBNs). NPBNs specify the nodes as arbitrary invertible distribution functions and the arcs as (conditional) rank correlations realized by a chosen one-parameter bivariate copula (Kurowicka and Cooke, 2005). This construction has two main implications: the parent-child dependence is realized by bivariate pieces of dependence, and the information required to quantify the network reduces to a number of marginal distributions equal to the number of nodes and a number of (conditional) dependence

parameters (parameterized by Spearman's rank correlation) equal to the number of arcs in the network (Hanea et al., 2015).

Hanea et al. (2015) demonstrated that the vine-copula based approach determines a unique joint distribution of the $n$ nodes given: (1) a DAG with $n$ nodes specifying the conditional independence relationships; (2) $n$ variables $\{X_1, \cdots, X_n\}$ assigned to the nodes and described by invertible marginal distributions $\{F_1, \cdots, F_n\}$; (3) arcs $i_{p-k} \to i$ for the node $i$ and its ordered set of $p$ parent nodes $S_i = \{i_1, \cdots, i_p\}$ specified by the (conditional) rank correlation in Eq. 2; (4) a copula realizing the (conditional) correlations in (3), for which correlation 0 entitles independence. It is worth noting that the parent set $S_i$ for node $i$ does not have a unique order.

$$\begin{cases} r_{i,i_{p-k}}, & k = 0 \\ r_{i,i_{p-k}|i_p,\cdots,i_{p-k+1}}, & 1 \leq k \leq p-1 \end{cases} \tag{2}$$

Considering the DAG in Figure 1, the joint probability of the associated NPBN is uniquely quantified given invertible marginal distributions $\{F_1, F_2, F_3\}$ and the (conditional) rank correlation for the two arcs $r_{1,3}$ and $r_{2,3|1}$, or $r_{2,3}$ and $r_{1,3|2}$ depending on the parent ordering for node 3.

The choice of the copula to quantify the arcs is arbitrary. However, only the joint normal copula allows rapid calculation and inference for complex problems (Hanea et al., 2015). For this reason, in this study, we adopt the protocol presented in Hanea et al. (2006) based on the Gaussian copula assumption. This protocol computes the joint distribution function of $n$ variables $\{X_1, \cdots, X_n\}$ with invertible marginal distributions $\{F_1, \cdots, F_n\}$ by:

- transforming the set of variables $X$ in standard normal variables $Y$ via the transformation $Y_i = \Phi^{-1}(F_i(X_i))$ for each node $i$, where $\Phi$ is the univariate standard normal distribution. The transformation is strictly increasing, so after the transformation the (conditional) rank correlation is unchanged;

- assigning to each arc of the network the quantity $\rho_{i,j|D} = 2\sin(\pi \cdot r_{j,i|D}/6)$, where $(i,j)$ and $D$ are respectively the conditioned and the conditioning set, $r_{j,i|D}$ and $\rho_{i,j|D}$ are respectively the conditional rank correlation and the partial product moment of the normal variables. A unique joint normal distribution, and so a unique correlation matrix, satisfying the partial correlation specification is determined;

- computing the correlation matrix $R$ recursively based on the partial correlations.

The joint distribution of the initial variables $X$ and their specified dependence is then realized by sampling from the joint normal distribution with correlation matrix $R$ a sample $\tilde{Y}$ and transforming it back to its original units via $\tilde{X}_i = F_i^{-1}(\Phi(\tilde{Y}_i))$ for every node $i$.

NPBNs based on the normal copula assumption are implemented in the open-source Matlab toolbox BANSHEE (Paprotny et al., 2020), which is used in this study to carry out the analyses.

## 4 NPBN as model for river discharge generation

The aim of this study is to investigate the suitability of NPBNs to reproduce catchment-scale hydrological dynamics. The rationale adopted to identify suitable DAG consists of representing a catchment as a system in which discharge is generated by the interaction between the input of the system, e.g., precipitation, the state of the system, e.g., soil moisture, and the output of the system, e.g., river discharge (Fig. 4a). This schematization allows us to define, via the associated NPBN, the joint probability distribution function of the variables (nodes) representing the input, state, and output of the system-catchment and subsequently infer the variable of interest, i.e., river discharge, via conditioning on the remaining variables. This schematization, hereafter graph type I, can easily be extended to include additional variables, such as physical attributes (e.g., elevation) of the system-catchment, resulting in the schematization in Figure 4b, hereafter graph type II. Graph type I determines the joint probability distribution of input (I), output (O), state (S) of a catchment considered as single element, i.e., the nodes are defined by the observations taken at one single catchment. Such joint distributions can be used to infer information on that single catchment. On the other hand, graph II defines the joint probability distribution of input (I), output (O), state (S) and attribute (A) of the catchments and the nodes are defined by pooled observations derived by merging observations at multiple locations. This way of defining the nodes implies that also the attribute nodes, which are constant value in time for a given catchment, become random variables and so they can be modeled as additional nodes in the network. From the joint distribution defined by the graph type II, we can derive the joint probability distribution of the input, output, and state variables of one single catchment, i.e., graph type I, via conditioning on the attributes of that catchment, $F_{g-I}(I,O,S) = F_{g-II}(I,O,S|A_{g-I})$. $F_{g-I}(I,O,S)$, derived from conditioning $F_{g-II}(I,O,S,A)$, benefits from information provided by similar catchments. Graph type II can be then implemented for ungauged catchments by exploiting information from gauged catchments with similar attributes.

In graph type I the following continuous hydro-meteorological variables will here be considered: $P_{max}$, $T$, $R$, $V_p$, (input), $Q_{max}$ (output), and $C_m$ (proxy for system state component). In graph type II the following nodes are added: $Elv$, $Slp$, $Ar$, $p_s$, and $ff$ (system attributes, Table 1). Note that remote sensing soil moisture data (ESA CCI soil moisture) were tested has variable representing the state of the system-catchment in a preliminary analysis (not shown here). However, missing values in soil moisture data, coarse spatial resolution, and the time lag between the response of river discharge and soil moisture to external input, such as precipitation, led us to rather use here monthly runoff coefficient as proxy for system-state.

Network selection, i.e., moving from a graph to a DAG by selecting arcs connecting a given set of nodes to model dependence, is challenging due to the high number of possible configurations describing a given set of variables. In this study, we a priori selected two DAGs: a DAG in which the variables are parent nodes with one child being the variable $Q_{max}$ and a DAG in which all the variables are connected via arcs resulting in a saturated network. We will refer to them as Unsaturated Network (UN) and Saturated Network (SN) respectively. UN can be considered as a multilinear regression function in which the discharge is the dependent variable and the remaining variables are the independent (explanatory) variables with coefficients defined by the rank correlation between the variables and discharge. Such explanatory variables are assumed to be independent of each other. However, in such a network, discharge is inferred as a function of all the other variables, while the other variables, e.g., $P_{max}$, $T$, $R$, $V_p$, and $C_m$, only depends on the discharge. This implies that this unsaturated network is suitable only if the variable

to be inferred is determined a priori, as in this case discharge, since our interest is in reproducing river discharge. SN, on the contrary, accounts for the interdependence of all the variables and does not have a pre-defined variable of interest which can influence the design of the network structure, as in UN. However, network selection, i.e., number and direction of the arcs and parent nodes ordering, is to some extent arbitrary. Besides, the strength of the arcs, determined by the dependence between nodes, can be based entirely on observations, as in this study, but can also be elicited from experts (Morales et al., 2008; Hanea et al., 2010). Hence, network definition and selection will be further discussed in the Discussion section.

In this study, we investigate two networks, one unsaturated (hereafter UN-1), as shown in Figure 4c, and one saturated (hereafter SN-1), as shown in Figure 4d, to generate river discharge considering each catchment as a single element.

To further explore the applicability of NPBNs in hydrologycal studies, we investigate the potential of a single saturated network (hereafter SN-C, Fig. 4e) to reproduce monthly maximum river discharge over many catchments and eventually also in ungauged basins. Such network builds upon network SN-1 and, in addition, includes attribute nodes. We implement SN-C network on a sub-sample of the 240 catchments having positive correlation between $P_{max}$ and $Q_{max}$ and Nash-Sutcliffe Efficiencies (NSE) $\geq 0.5$, calculated using the network SN-1. In doing so, we a priori group catchments with a similar property and performance at the catchment level. From the network SN-C, we only infer statistical characteristics of river discharge rather than specific events, as we do from UN-1 an SN-1.

The joint distribution function associated to each network (UN-1, SN-1, and SN-C) is derived following the protocol presented in Hanea et al. (2006) and discussed in the previous section. We assume a normal copula for quantifying (conditional) rank correlations and empirical cumulative distribution functions for describing the marginal distributions of the different nodes.

## 4.1  NPBN testing

To assess the potential of NPBNs as probabilistic models for catchment dynamics, we first test the networks (UN-1, SN-1, and SN-C) as descriptive models. Subsequently, we evaluate the networks as predictive models. In this study, the term *testing process* refers to analyses performed on the descriptive models, while the term *evaluation process* refers to analyses performed on predictive models. In the evaluation process, elsewhere also referred to as validation process, the data set used to determine the networks, i.e. quantification of the dependence between nodes, differs from the data set used to evaluate the performance of the network in estimating discharge. In the testing process, elsewhere also referred to as verification process (Hanea et al., 2015), the entire set of observations available is used to first determine the networks and then to test it via diagnostic metrics, such as here NSE and KS test. This approach implies that the minimum requirement for a network is to reproduce the observations used for quantifying the model itself. At the same time, it can happen that the limited amount of available observations does not allow the definition of a representative train- and a test-set (Hanea et al., 2015), preventing the possibility of evaluating the predictive capabilities of the model. In the evaluation process, we perform a k-fold cross validation by random selecting 10 years, between 1980 and 2013, as test-set while the remaining years are used as training-set.

In both testing and evaluation process, we first test the assumption of the joint normal copula for modelling the bivariate dependence via the Cramer-Von-Mises test. Then, we use the $d - calibration$ score to test the assumption that the network

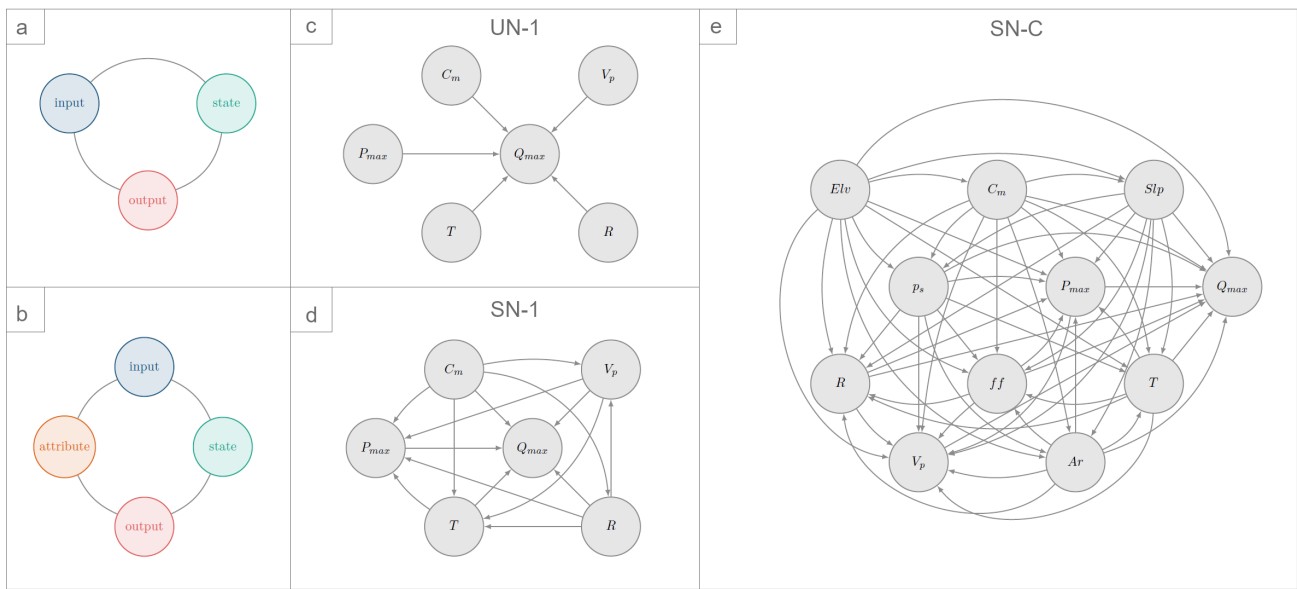

**Figure 4.** Graphs and qualitative networks (DAG) used in this study. Panels (a) graph type I and (b) graph type II show the rationale underlying the selection of the variables in the networks. Panels (c) and (d) represents the networks for analysing catchments as single elements, UN-1 and SN-1 respectively. Panel (e) shows the network for analysing a group of catchments from contrasting climate, SN-C.

selected, UN-1, SN-1, or SN-C, can model the overall multivariate dependence structure. Afterwards, we test and evaluate the performances of the networks as descriptive and predictive models, respectively, in inferring discharge data via the two-sample Kormogorov-Smirnov test and the Nash-Sutcliffe Efficiency (NSE) coefficient.

The Cramer-Von-Mises (CvM) statistic $S$ provides an indication of the distance between the empirical copula $C_n$ and the theoretical copula $C_\theta$, e.g. Gaussian copula (Genest and Favre, 2007):

$$S = n \sum_{i=1}^{n} \left\{ C_n \left( \frac{R_i^1}{n+1}, \frac{R_i^2}{n+1} \right) - C_\theta \left( \frac{R_i^1}{n+1}, \frac{R_i^2}{n+1} \right) \right\}^2 \tag{3}$$

where $R_i^1$ and $R_i^2$ are the $i^{th}$ ranks of the $n$ observations. CvM test provides a measure of goodness-of-fit of a theoretical
copula. $S = 0$ means perfect fit. We test the normal copula assumption in modelling dependence via bivariate correlation by performing the CvM test on the pairs of variables resulting from the combination of the network nodes (variables). We compare the empirical copula of each pair with 4 different parametric copulas, widely used in hydrological studies, namely: Gaussian (or Normal), Frank, Gumbel, and Clayton. The characteristics of the Gaussian and the Frank copula are similar, in the sense that they are both suitable models for variables which do not have strong association between each other when both take low/high
value. On the other hand, Gumbel and Clayton copulas are suited to model variables with a strong dependence at upper and

lower tail, respectively.

The $d - calibration$ $(d_c)$ metric (Morales-Nápoles et al., 2014b) proves a goodness-of-fit measure of the joint probability distribution function defined via NPBN against the empirical distribution.

$$d_c = 1 - d_H \tag{4}$$

where $d_h$ is the Heillinger distance between the empirical correlation matrix of the variables (nodes of the network) and the NPBN correlation matrix. $d_c$ takes values between 0 and 1, with high score implying that the two correlation matrices are similar.

The two-sample Kolmogorov-Smirnov (KS) is a non-parametric hypothesis testing technique assessing whether two samples, $Y$ and $\tilde{Y}$, belong to the same population (Massey, 1951). The KS test statistic $D^*$ is defined as:

$$D^* = \max_y \left( |F_Y(y) - F_{\tilde{Y}}(y)| \right) \tag{5}$$

The null-hypothesis $H_0$ is $F_Y = F_{\tilde{Y}}$ against alternatives. In this study, we consider a level of significance $\alpha = 0.05$.

The Nash–Sutcliffe Efficiency coefficient (NSE) (Nash and Sutcliffe, 1970) measures the predictive capabilities of the NPBN.

$$NSE = 1 - \frac{\sum_{i=1}^{N}(y_{sim}^i - y_{obs}^i)}{\sum_{i=1}^{N}(y_{obs}^i - \bar{y}_{obs})} \tag{6}$$

where $y_{sim}$ is the simulated specific discharge, $y_{obs}$ is the observed specific discharge, $\bar{y}_{obs}$ is the observations mean, and $N$ is the total number of observations. Values of NSE lower than 0 indicate that the observations mean ($\bar{y}_{obs}$) is a better predictor than the model adopted. Values close to 1 suggest very good model performances.

NPBN treats hydro-meteorological data and catchment attributes as random variables. This implies that during the inference process, the NPBN returns, at each time step, a conditional distribution function of the target variable, i.e., the distribution of maximum monthly river discharge conditioned on the remaining hydro-meteorological data and attributes. From this conditional distribution of river discharge, 1000 possible discharge realization are sampled and the $50^{th}$ percentile is taken as the estimated discharge value for that particular combination of hydro-meteorological data and attributes. Similarly, the confidence interval (CI) of the estimated discharge value is determined as the $5^{th}$ and the $95^{th}$ percentile of the 1000 realizations of the conditional distribution.

## 5 Results

In this section, we first show the potential of NPBNs in estimating monthly maximum river discharge when a catchment is modeled as single element. Afterwards, we present the capability of NPBNs to model catchments in a cluster to eventually infer river discharge of an ungauged basin given its attributes.

### 5.1 Catchment as single element

We first analyse the performances of the networks UN-1 and SN-1 as descriptive models. In Figure 5a, the results of the CvM test show that the best copula model among the four tested is the Frank copula for $\sim 55\%$ of the pairs, the Gaussian copula for $\sim 10\%$, and the Gumbel and Clayton for $\sim 20\%$ of the pairs respectively. This suggests that about 65% of the pairs, i.e. pairs best modeled with either Frank or Gaussian copula, show a dependence without a strong association between low/high value. Hence, this result supports the normal copula assumption of NPBNs, since the Gaussian copula is a suitable model for such type of dependence. In Figure 5b, boxplots summarizing the results in terms of $d-calibration$ score indicate that, on average, network SN-1, with a median of $\sim 0.8$, better capture the overall dependence between variables. Indeed, the $d-calibration$ score compares the empirical correlation matrix of the variables with the correlation matrix resulting from the DAG. Low $d-calibration$ score for network UN-1 (median of $\sim 0.25$) can be linked to the strong assumption of independence between pairs of variables in which one variable in the pair is not discharge. A further insight about the suitability of networks UN-1 and SN-1 is via NSE, which describes how a network is able to reproduce discharge events given information about $P_{max}$, $T$, $R$, $V_p$, and $C_m$. For catchments in which the correlation between $Q_{max}$ and $P_{max}$ is negative, i.e. catchments in water-limited regions and at high elevations, network SN-1 returns higher value of NSE compared to network UN-1 (red dots above the identity line, Fig. 5c). This result provides evidence that, in catchments where the discharge generation process is not predominantly precipitation driven, it is important to account for the interaction between other hydro-meteorological variables and catchment current state. Finally, based on the expected $Q_{max}$ simulated with the network, for each catchment, we estimated the $0.5$, $0.05$, and $0.95$-quantile and we compare them with the same quantiles from observations. While the observed and simulated mid-quantiles in both networks UN-1 and SN-1, respectively, broadly correspond (Fig. 5d), Figure 5e shows that in network SN-1 lower quantiles are overestimated (dark grey histogram with most mass on values $> 1$), while upper quantiles are underestimated (dark grey histogram with most mass on values $< 1$), Fig. 5f. On the contrary, the network UN-1 shows greater variability in simulating discharge since a clear pattern of over- or under-estimation is not visible especially for the $0.95-$quantiles, Fig. 5f . These results likely reflect the property of the Gaussian copula of no tail dependence.

The preliminary analysis on the descriptive capabilities of networks UN-1 and SN-1 suggests that the network SN-1 is better suited for describing the dynamics of river discharge compared to UN-1. However, when we look more in depth into network SN-1 performances, we can observe that only 66% of the catchments have a NSE higher than $0.5$ (Fig. 6a), which in the literature is considered as acceptable performing model (Moriasi et al., 2007; Newman et al., 2015). Such catchments receive precipitation mainly in winter (mean $p_s$ $\sim$-0.19), are located in energy-limited regions (mean $Ar$ $\sim$0.78), and are mostly green areas (mean fraction of forest $\sim 0.91$). At the same time, in 85% of the catchments the $H_0$ of the KS test cannot be rejected

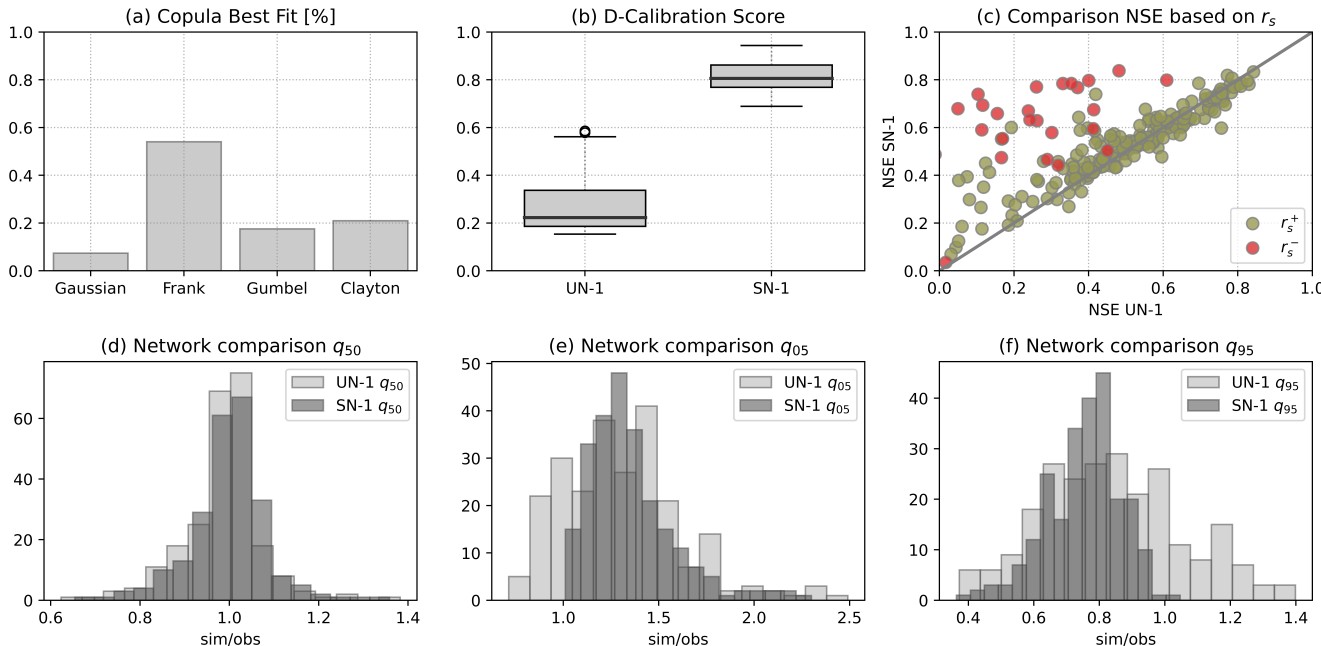

**Figure 5.** Results of the testing process when the networks UN-1 and SN-1 are used as descriptive models of the 240 catchments considered as single elements. Panel (a) shows the percentage of pairs having the copula on the x-axis as best-fit. Panel (b) shows the variability of the $d-calibration$ score across catchments. Panel (c) shows the comparison between UN-1 and SN-1 in terms of NSE as a function of the sign of the correlation ($r_s$) between $Q_{max}$ and $P_{max}$. Red dots indicate negative correlation ($r_s^-$), while green dots ($r_s^+$) indicate positive correlation. The identity line (grey) is used as indicator to visually compare the results from UN-1 and SN-1 networks. Dots on the line indicate matching results between the two networks, while dots above/below the line indicate higher values of NSE associated to NS-1/UN-1 network. Panels (d) to (f) show the histograms of the ratio between simulated and observed discharge quantiles, i.e., 0.5, 0.05, and 0.95-quantiles, for the network UN-1 (light grey) and SN-1 (dark grey). Panel (d) shows that the networks UN-1 and SN-1 simulate the 0.5-quantile similarly, while panel (e) and (f) show that the network SN-1 generally over- and under-estimate the $0.05-$ and the $0.95-$ quantile respectively. Panel (e) and (f) show also that the network UN-1 does not have a clear tendency to over- or under-estimate the quantiles.

(average $p_{value}$ = 0.49), suggesting that the sample of monthly maximum river discharge simulated from the network derives from the same distribution as the sample observed. This result implies that the network captures well the average behaviour of the catchment in the long term (tested via KS test), while has limitations when inferring single events (tested via NSE).

To further investigate the ability of NPBNs to estimate monthly maximum river discharge, we evaluate the performances of the network SN-1 as predictive model. We limit the investigation to the network SN-1 since the above results suggest that it is a descriptive model for a larger number of catchments with contrasting characteristics compared to network UN-1. The k-fold cross validation test is applied to catchments having NSE greater than 0.5 in the testing process described above, here 159 catchments. We perform 5 simulation runs and, in every run, 10 years where randomly selected as test-set. We consider

the performances in terms of NSE calculated as mean value of the 5 runs. Results indicate that $25\%$ of the catchments have NSE of the test-set higher than the training-set (Fig. 6a). In general, one would expect a better performance in the test-set compared to the training set, since the metric for evaluating model performances uses the same data set for quantifying and testing the model. Hence, the fact that the training-set performs better than the test-test could depend on the random selection of the years for evaluating the network. This random procedure might have split the original data set into two data set with different characteristics. This could be due to the relatively small number of years of observations which are subsequently divided in even smaller data sets. In contrast, around $55\%$ of the catchments analysed (about $40\%$ of the total catchments) have a NSE in the test-set $\geq 0.5$ (Fig. 6b) and, at the same time, a NSE in the test-set equal or lower than the training-set (Fig. 6a), meaning that network SN-1 in these catchments provides reliable estimates of river discharge events and long term characteristics. In a recent study, Ren et al. (2020) investigated the performances of regression models based on a variety of filter-based feature selection methods to estimate average monthly river discharge in three catchments from the CAMELS data set. The results obtained in terms of NSE ranged from $\sim$0.6 to $\sim$0.8, values similar to average (mean) performance of the network SN-1 (NSE $\sim$0.596) here investigated for maximum river discharge. Kratzert et al. (2019) used CAMELS data set to evaluate the performances of hydrological models. They investigated the performances of Long Short Term Memory (LSTM) network to estimate daily river discharge in 530 catchments and included also, among other models, the performances of the Sacramento Soil Moisture Accounting (SAC-SMA) conceptual model. For the sake of discussion, we look at the performances of LSTM network without catchments attributes and SAC-SMA from Kratzert et al. (2019) for a subset of catchments also analysed in this study. Kratzert et al. (2019) results are available at https://github.com/kratzert/lstm_for_pub. The LSTM network without catchments attributes and the SAC-SMA conceptual model for daily river discharge have an average (mean) performance of NSE $\sim$0.603 and $\sim$0.598 respectively. SN-1 network, here investigated, for monthly maximum river discharge has an average (mean) performance of NSE $\sim$0.596. In general, NSEs obtained for simulations on a daily temporal scale tend to be lower than the ones on a monthly temporal scale due to the higher amount of observations over a common fixed period of time (Moriasi et al., 2007). However, other studies suggest that for both daily and monthly model simulations a satisfactory performance is given when 0.37<NSE<0.75 (Van Liew et al., 2007). To further evaluate the performance of NPBNs for maximum river discharge, we perform the $KS$ test. In about $95\%$ of the catchment the $KS$ test $H_0$ cannot be rejected, Fig. 6(c). Such result is in agreement with the one found in the testing process of the descriptive model, being that the SN-1 network shows limitations when inferring single events (tested via NSE) but is fairly good when inferring long-term behaviour (tested via KS).

NPBNs provides a quantification of the uncertainty around the estimated river discharge values. We then quantify the uncertainty of the estimated maximum river discharge. On average and across all catchments, observed discharge in the test-set falls within the simulated confidence interval ($5^{th}$ and $95^{th}$ percentile) about $63\%$ of time, ranging between a minimum of 45 and max of $78\%$, Fig. 6(d).

To further evaluate the results of the SN-1 network in estimating monthly maximum river discharge, the hydrograph of three stations, i.e. #6746095 (Colorado), #11481200 (California), and #14306340 (Oregon), with contrasting characteristics are shown in Figure 7.

The catchment in Colorado is located in an water-limited area ($Ar \sim 1.1$) above 3000 m a.s.l., has a negative correlation be-

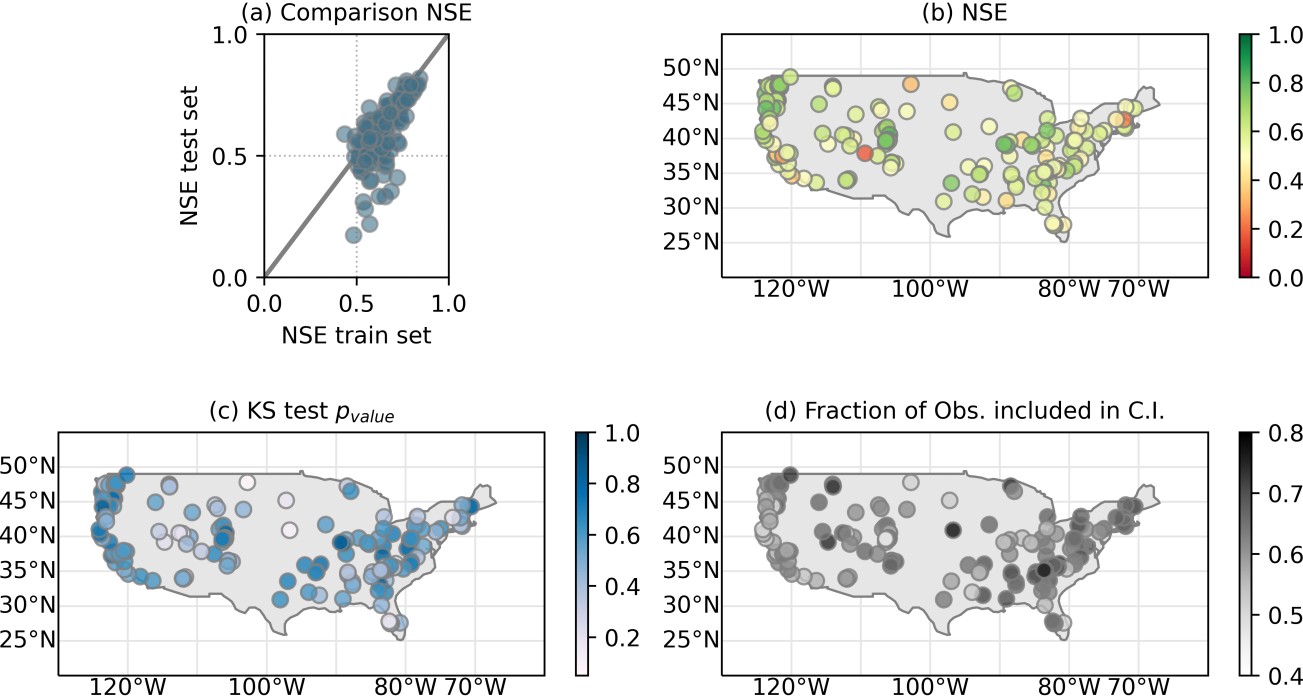

**Figure 6.** Performances of the SN-1 network in terms of NSE for the 159 catchments having NSE ≥ 0.5 in the testing process. Panel (a) compares the performances of the training- and test-set. Panel (b) shows the value of NSE per catchment. Panel (c) shows the $p_{value}$ resulted from the KS-test per catchment. Panel (d) shows the overall fraction of observations falling inside the estimated discharge uncertainty bounds per catchment.

tween $Q_{max}$ and $P_{max}$ and precipitation falls mainly in winter (negative value of $p_s$). The results of the evaluation process
show that network SN-1 can reproduce the statistical characteristics of the maximum river discharge observed (KS-$H_0$ non-rejected, $p_{value}$=0.82) as well as the seasonal variability (Fig. 7d). Moreover, the scatter plot in Figure 7a shows simulations in agreement with observations (Mean Absolute Percentage Error - MAPE - ∼0.47). The mean value of NSE across the 5 runs is 0.82 and 56% of the observations fall within the simulation confidence interval (Fig. 7d). The catchment in California is located in an energy-limited region ($Ar \sim 0.54$) at ∼ 300 m. a.s.l.. Here, there is a positive correlation between $Q_{max}$ and $P_{max}$ and
precipitation falls mainly in the winter season (negative $p_s$). Network SN-1 is able to simulate the statistical characteristics of the observed discharge (KS-$H_0$ non-rejected, $p_{value}$=0.72). Moreover, the simulations follow the seasonal variability of the observations (Fig. 7e), even though it is less pronounced than in Colorado, and ∼ 50 % of the observations fall within the model CI. The mean value of NSE across the 5 runs is 0.68 and the MAPE is ∼0.83. This result reflects the fact that few simulations in the test set (Fig. 7b) deviates significantly from observations. Finally, the catchment in Oregon is located in an
energy-limited region ($Ar \sim 0.38$) at ∼ 400 m. a.s.l. and it has a positive correlation between $Q_{max}$ and $P_{max}$. Precipitation falls mainly in the winter season (negative $p_s$). Network SN-1 is able to reproduce the statistical characteristics of maximum

river discharge (KS-$H_0$ non-rejected, $p_{value}$=0.86) and the discharge seasonal variability is captured by the model (Fig. 7f). The NSE across the 5 runs is 0.79 and MAPE is ~0.60. In the hydrograph in Figure 7f it can be observed that in 2001 the seasonal variability typical of the other years is less pronounced. Also, the CI (shaded red area) is larger compared to the rests. This is likely a consequence of the fact that three consecutive years (1999, 2000 and 2001, Fig. 7f) were randomly selected for model evaluation including the year 2001.

These results show the SN-1 network potential to model river discharge generation process in catchments with contrasting climate exploiting information from the interaction between the different inputs of the system-catchment, i.e., $R$, $V_p$, $T$, $P$, $C_m$, even when precipitation is not the main discharge driver, e.g., Colorado.

## 5.2 Catchments in cluster

We implement network SN-C on a sub-sample of 133 catchments having positive correlation between $P_{max}$ and $Q_{max}$ and NSE calculated based on network SN-1 greater than 0.5 in the previous analysis considering catchments as a single elements. We first test the performance of network SN-C as descriptive model. Similar to the results obtained previously, Frank and Gumbel copula are the best theoretical copulas for about $50\%$ of the pairs, supporting the choice of the NPBN. The $d - calibration$ score is about $0.84$, meaning that the network well captures the interdependence between variables obtained via empirical correlation matrix. In contrast, the KS test indicates that in only $20\%$ of the cases analysed here the model can reproduce monthly maximum river characteristics ($H_0$ cannot be rejected, average $p_{value}$0.24). Given the limitation of the descriptive model in reproducing statistical characteristics of maximum river discharge, single events are not inferred as for the networks UN-1 and SN-1.

We note that removing one station from the overall pooled of observations has very small effect on the empirical correlation matrix of the empirical variables, the correlation matrix associated with the network, and the cumulative distribution of each node: the observations belonging to one catchment are around $0.8\%$ of the total observations from all the catchments. This shows that network SN-C is quite robust. Hence, we further evaluate the robustness of network SN-C performances as predictive model by leave-one-out cross validation. The KS test is performed for each catchment using the value of monthly maximum river discharge observed and simulated via network SN-C calibrated without the information of the catchment analysed (evaluation process). This is done to assess the potential of such network in exploiting the information from catchments with similar attributes. The descriptive and the predictive models perform similarly, suggesting that network SN-C is quite robust. The KS test results show that in only $15\%$ of the sub-sample of catchments here analysed (Fig.e 8a green dots; $10\%$ of the total number of catchments) the $H_0$ cannot be rejected ($p_{value}$ 0.20) meaning that in only $15\%$ of the catchments the distribution of $Q_{max}$ simulated and $Q_{max}$ observed belong to the same distribution family. Such catchments are characterized by a relatively strong correlation between $P_{max}$ and $Q_{max}$ (median around 0.53), are in energy-limited regions (aridity median $\sim 0.74$) at moderate elevations (median $\sim$500 m.a.s.l.). Moreover, in such catchments precipitation is on average constant over the year ($p_s$ 0.08). However, there is no clear pattern in catchment attributes of those catchments with $H_0$ rejected in the predictive model but not rejected in the descriptive model.

To further analyse the results, we look at one catchment in California (#11481200), where the $H_0$ is rejected, and one in

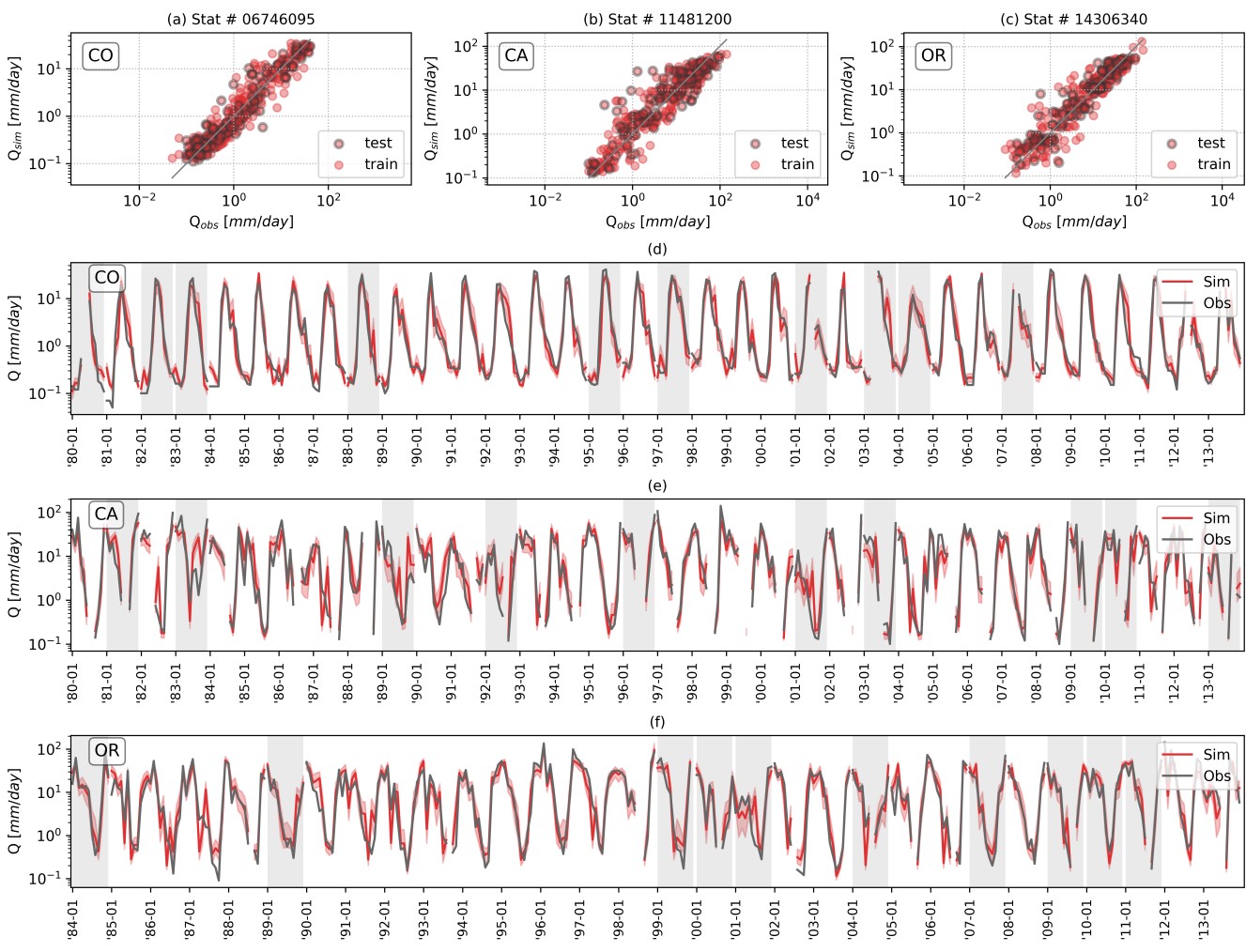

**Figure 7.** Comparison between monthly maximum river discharge simulations from run 1 and observations of three different catchments. The grey shaded areas indicate years belonging to the test-set. The shaded red areas represent the simulation confidence interval evaluated as the $5^{th}$ and the $95^{th}$ percentile.

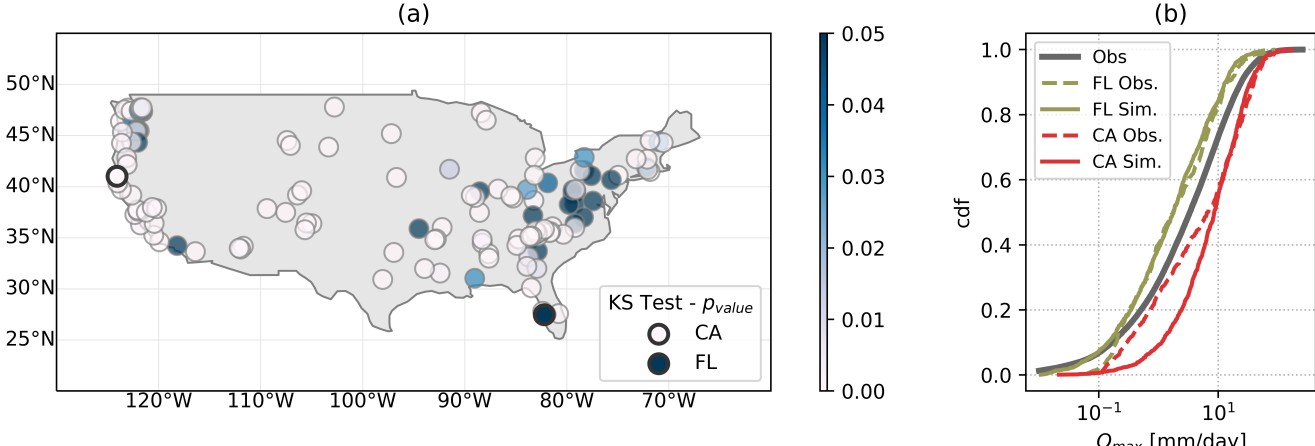

**Figure 8.** Results from the KS test on SN-C network. Panel (a) shoes the $p_{value}$ of the KS test results at the corresponding catchment location. $p_{value} \leq 0.05$ indicates that $H_0$ is rejected. Two locations are highlighted: station #11481200 in California and station #02299950 in Florida. Such stations are further analyzed in panel (b). Here, the grey solid line indicate the cumulative distribution function modelling the node $Q_{max}$ in the network SN-C. The red lines represent the observed (dashed) and simulated (solid line) $Q_{max}$ distribution at the catchment in California. The green lines represent the observed (dashed line) and simulated (solid line) $Q_{max}$ distribution at the catchment in Florida. The comparison of the colored lines with the grey line shows that by conditioning on catchment's attributes, the SN-C network returns discharge values in the range of the discharge observable at the catchment. The comparison between solid and dashed lines of the same color indicates whether conditioning on attributes is sufficient for a good discharge estimation. In Florida the model performs well (the two lines overlap), while in California the model cannot well reproduce low quantiles.

Florida (#02299950) where the $H_0$ cannot be rejected. Figure 8b shows that conditioning network SN-C on catchments attributes leads to a sub-sample of discharge values in the range of the observed ones. However, low quantiles are not well captured (dashed colored lines departing from the corresponding solid lines in Figure 8(b)), especially in the catchment in California (red line).

## 6 Discussion and Challenges

The performances of NPBNs indicate that the interdependence between hydro-meteorological information should be explicitly modelled to better capture the river discharge characteristics at the catchment level: network SN-1 provides higher NSE values compared to network UN-1. Additionally, it suggests that at least the networks trained at the catchment scale, i.e. SN-1 and UN-1, show potential to describe the hydrological response, while more research will be needed to develop meaningful NPBNs trained across a range of multiple catchments. Indeed, in our study, river discharge could only be poorly captured by this network type, i.e. SN-C, which reflects the common issue of information transfer in hydrological modelling.

By further considering the results just obtained we identify six issues related to both NPBNs properties (i.e., data quality and quantity, independence of weather events, feasibility of testing procedure, and Gaussian-copula assumption) and river discharge generating process (i.e., catchment heterogeneity, and interacting spatial and temporal scales) that we believe have influenced the networks' performances. We discuss this issues in details and, when possible, we propose a way to address them.

*Catchment heterogeneity.* River discharge generation is the result of underlying physical processes at different time scales and it occurs in catchments with spatially heterogeneous characteristics. Such complexity affects the performances of a fixed model configuration (i.e., network nodes and interdependence as summarized by arcs) in which the time scale of the different processes involved is implicitly treated.

*Data quality and quantity.* NPBNs, similarly to other models, are sensitive to the quantity and quality of the data used for network quantification. In a NPBN, hydro-meteorological observations and catchment attributes are modelled as random variables, via parametric (or empirical) distribution function learned from the data themselves. This requires that (for static models) observations used in the training process are representative also for future inference, i.e., they have time-invariant statistics. Such statistics are quite sensitive to the quantity and quality (measurement errors) of the data. This is particularly relevant when modelling extremes, both low and high, since observations of which are already scarce. The results obtained in this study reflect to some extent such difficulty. For example, network SN-1 for modelling a catchment as single element over- and under-estimate the $5^{th}$ and the $95^{th}$ percentile respectively (Fig. 5e-f).

*Interacting spatial and temporal scales.* River discharge at the outlet of a catchment is generated from the interaction between many, partially simultaneously occurring physical processes, such as direct runoff, infiltration, and evaporation. These processes are characterized by different spatial and temporal scales and can vary substantially within and across catchments. Specifically, as tested here, in a NPBN the causal relationship between river discharge and hydro-meteorological variables is modeled via (conditional) correlation, which, however, is a measure of dependence and does not imply causation. Therefore, to model the temporal component of the underlying physical processes, we sampled hydro-meteorological variables within a 7-day time window prior to maximum discharge event. However, with this procedure we might have missed some relevant interaction, such as the different response of river discharge to a precipitation event due to soil conditions. In this regard, further analysis, for example, on how to account more explicitly for soil moisture content (here we only considered monthly runoff coefficient) could improve the results. Our preliminary assessment (results are not shown here) is that available remote sensing soil moisture data are not enough to provide a representative multivariate data set because of high amounts of missing data, especially over the winter season. Furthermore, the variability of soil moisture content is much higher than discharge, for example, in response to a precipitation event. This is also due to the fact that soil moisture is more sensitive to other input variables, such as temperature, compared to river discharge. Hence, it is challenging to identify at which time frame (i.e., maximum/mean over week/day) information on soil moisture are relevant for improving maximum river discharge generation at monthly scale.

*Independence of weather events.* NPBNs are graphical models to construct a joint distribution function on a given set of random variables represented as nodes in a DAG. At a monthly time scale, the temporal scale considered in this study, samples used in the quantification process are not always time-independent. The sampling procedure of the multivariate data set based on monthly maximum events contribute to guarantee the time-independence property of the events sampled, since events should be driven by different weather systems. However, some autocorrelation, particularly in discharge data was observed (Fig. S3). To address this, future research exploring Dynamic Non-Parametric Bayesian Networks, is recommended Hanea et al. (2013).

*Feasibility of testing procedure.* BNs model selection is a challenging task due to the high number of possible DAG configurations determining a multivariate probability function describing a given set of variables, where each configuration is de facto a possible hypothesis on the system functioning and may in principle be tested. Furthermore, the same DAG can be quantified differently based on the ordering of the parent nodes. NPBNs specify the nodes as arbitrary invertible distribution functions and the arcs as (conditional) rank correlation (Kurowicka and Cooke, 2005). The conditional correlation depends on the parents ordering chosen for a given child (node). For example, the network in Figure 3 can be quantified by two pairs of (conditional) rank correlations: $r_{1,3}$ and $r_{2,3|1}$ ; or $r_{2,3}$ and $r_{1,3|2}$. In the former case the parent order is $\{1,2\}$, in the latter $\{2,1\}$. In general, given $n$ nodes, the saturated DAG (all the nodes connected) has $n^{n-2}$ possible parent-child combinations (Morales-Napoles, 2010), and this number increases when testing other DAG configurations justified by prior information. This large number of potential models would render network selection on the bases of a "brute force" procedure (evaluating a large portion of the space of models) computationally unfeasible. For such reason, we imposed the network configuration based on prior knowledge about the relationship between the variables, and we investigated model performances based on the model outcome. This strategy, however, can affect the capability of the model as catchment descriptor and can conceal relationships that a priori may seem illogical or unlikely. Hydrological applications require a good knowledge of the interactions and dependencies in a system, which are often largely unknown beyond individual catchments and this is reflected in the fact that in this study NPBNs, which requires information to model dependence, perform better for catchments as single elements than for catchments in cluster.

*Gaussian-copula assumption.* NPBNs, introduced by Kurowicka and Cooke (2005) and implemented in this study, assume that the arcs are quantified via the normal or Gaussian copula (Nelsen, 2006), because only this copula allows rapid calculation and inference for complex problems (Hanea et al., 2015). However, the normal copula does not capture important asymmetries often observed in data (for example, lower and upper tail dependence), meaning that it is not able to properly model relationships where extreme values (minimum and/or maximum values) are more strongly associated than values not in the joint tails of the distribution. This issue can be solved by quantifying the arcs based on a different copula family. In this way, the join distribution function of the nodes in the network is realized via vine-copulas. However, a complete theory of vine-copulas conditionalization does not exist, making the process at the least computationally demanding and consequently preventing their

applicability to high-dimensional studies such this one.

# 7 Conclusions

The main objective of this study was to further explore and test the suitability of NPBNs as a tool to reproduce catchment-scale hydrological dynamics and to explore challenges involved when inferring monthly maximum discharge, since applications of
510 NPBNs in hydrology are still limited. In this study, we investigated 240 catchments across the United States, obtained from CAMELS data set, aiming at testing the ability of NPBNs to estimate monthly maximum river discharge. We showed that, once a NPBN is defined, it is straightforward to infer any of its variables, i.e. discharge, when the remaining variables are known, and extend the network itself with additional variables, i.e. going from network SN-1 containing only hydro-meteorological variables to network SN-C containing hydro-meteorological variables and catchments' attributes. The NPBNs individually trained
to specific catchments showed potential to reproduce monthly maximum river discharge in a wide range of environments with an average NSE of 0.59 (predictive models), while in the literature the performance of regression models for average monthly river discharge were NSE ∼0.6 to ∼0.8 (Ren et al., 2020), and the performances for daily river discharge were NSE of ∼0.603 and ∼0.598 for LSTM network and SAC-SMA model respectively (Kratzert et al., 2019). On the other hand, the network SN-C trained across sets of many contrasting catchments exhibited modest skill, i.e., only 10% of the catchments with an average
KS test-$p_{value}$ of 0.20. This calls for additional analyses to overcome the limitations encountered and discussed in the previous section to support future studies using statistical based models. Future research directions will focus on improving the understanding of the time scale at which the many hydro-meteorological variables leading to discharge generation interact. For this purpose we recommend investigating the potential of dynamic BNs to explicitly model the "memory" of the system (i.e. autocorrelation in the variables). Another research direction is exploring vine-copulas to better capture the possible asymmetries
observed in extremes.

*Data availability.* The data used in this study are from the CAMELS project and can be found at https://ral.ucar.edu/solutions/products/camels.

*Code and data availability.* NPBNs were modeled using the Matlab toolbox BANSHEE https://github.com/dompap/BANSHEE.

*Author contributions.* ER, MH, and OMN developed the study. ER carried out the numerical analyses and prepared the manuscript preliminary draft. MH and OMN contributed to the final version of the manuscript and the discussion of the results.

*Competing interests.* Markus Hrachowitz is editor of HESS.

*Disclaimer.* TEXT

*Acknowledgements.* This project has received funding from the European Union's Horizon 2020 research and innovation programme under the Marie Skłodowska-Curie grant agreement No 707404

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
