# Peer review of "Applying Non-Parametric Bayesian Network to estimate monthly maximum river discharge: potential and challenges"

_Hydrology and Earth System Sciences, 2021_

## Author Response (AR1)

**Responses to Reviewers**
*Manuscript hess-2021-229: "Applying Non-Parametric Bayesian Network to estimate monthly maximum river discharge: potential and challenges" Elisa Ragno et al.*

Dear Editor,

Thank you very much for considering our manuscript for publication and for taking the time to review our work in detail. Below the answers to the comments provided by the reviewers in blue.

Best,

E. Ragno, on behalf of M. Hrachowitz, and O. Morales-Nápoles

**Reviewer 1:**

In the study, the authors mainly investigated the performance of the Non-Parametric Bayesian Network for the estimation of monthly maximum river discharge, and also discussed its challenges, with a case study in the 240 catchments in USA. Overall, the paper was rewritten well, and many details were clearly explained. However, there two main issues that should be clarified to further improve the quality of the paper before its submission.

First, the authors briefly explained the motivation of this study as: "very little attention has so far been given to explicitly representing the interdependence between inflow and outflow via probability functions". However, it is not clear enough, as there have been many methods used for describing the relationship among variables through probability functions. The key issue should be further explained very clearly to clarify the potential novelty of this study in Introduction.

We would like to thank the Reviewer for taking the time to review our work in detail. Non-parametric Bayesian Networks are tools for defining the joint distribution function of a set of variables. This joint distribution may be used to generate river discharge samples (or samples of any other variable in the model if required). The advantage of such a method is that the joint probability distribution is determined by defining the dependence between pair of variables. Such a non-parametric joint probability distribution is then more flexible compared to a theoretical parametric multivariate distribution because the dependence between variables is not fixed by the theoretical parametric model, but it depends on how the variables (nodes of the network) are connected to each other (arcs and parenting order). The dependence between pairs of variables can be determined based on prior knowledge of the underlying system dynamics, so it can be case dependent. As the Reviewer mentioned, other studies have implemented such methods, and references are provided in the manuscript – but applications in hydrology remain scarce. Here, we further explore the potential of such methods in providing estimates of river discharge by defining the joint distribution between environmental variables that are used for physical-based hydrological models and which are considered drivers of the discharge generation process. Then, we use the joint probability distribution to derive discharge via conditional probability, rather than characterizing the joint occurrence of the modelled variables, which is the most common implementation of multivariate distribution function.

In the revised manuscript, we have modified the *Introduction* to better highlight the novelty of the study and the main objective. Please refer to Lines 44-79 of the revised manuscript.

Second, there lacks "comparison discussion" between the NPBN-based results here and the previous studies in the study area. There have been so many studies in these catchments and others in USA. Without comparison, the advantages and challenges of the NPBN model cannot be easily understood. Thus I suggest adding some comparison contents to prove the advantages of the NPBN.

We agree with the Reviewer's comment that the lack of comparison of the NPBN-based results and previous studies makes difficult to appreciate the advantages and challenges of the proposed model. We would like to add also that the aim of this study was to provide a comprehensive analysis of the suitability of Non-Parametric Bayesian Networks (NPBNs) to derive river discharge via conditional probability given its several advantages in terms of the characteristics of NPBNs compared to other process-agnostic models.

Following the Reviewer suggestions, in the revise manuscript, we included the performances of other methods used for river discharge generation found in the literature

> Lines 353-368 – "*In a recent study, Ren et al. (2020) investigated the performances of regression models based on a variety of filter-based feature selection methods to estimate average monthly river discharge in three catchments from the CAMELS data set. The results obtained in terms of NSE ranged from ~0.6 to ~0.8, values similar to average (mean) performance of the network SN-1 (NSE ~0.596) here investigated for maximum river discharge. Kratzert et al. (2019) used CAMELS data set to evaluate the performances of hydrological models. They investigated the performances of Long Short Term Memory (LSTM) network to estimate daily river discharge in 530 catchments and included also, among other models, the performances of the Sacramento Soil Moisture Accounting (SAC-SMA) conceptual model. For the sake of discussion, we look at the performances of LSTM network without catchments attributes and SAC-SMA from Kratzert et al. (2019) for a subset of catchments also analysed in this study. Kratzert et al. (2019) results are available at https://github.com/kratzert/lstm_for_pub. The LSTM network without catchments attributes and the SAC-SMA conceptual model for daily river discharge have an average (mean) performance of NSE ~ 0.603 and 0.598 respectively. SN-1 network, here investigated, for monthly maximum river discharge has an average (mean) performance of NSE ~0.596. In general, NSEs obtained for simulations on a daily temporal scale tend to be lower than the ones on a monthly temporal scale due to the higher amount of observations over a common fixed period of time (Moriasi et al., 2007). However, other studies suggest that for both daily and monthly model simulations a satisfactory performance is given when 0.37<NSE<0.75 (Van Liew et al., 2007). "*

> Lines 514-518 – Conclusion: "*The NPBNs individually trained to specific catchments showed potential to reproduce monthly maximum river discharge in a wide range of environments with an average NSE of 0.59 (predictive models), while in the literature the performance of regression models for average monthly river discharge were NSE ~0.6 to ~0.8 (Ren et al., 2020), and the performances for daily river discharge were NSE of ~0.603 and ~0.598 for LSTM network and SAC-SMA model respectively (Kratzert et al., 2019) "*

**Reviewer #2**

This manuscript explores the application of a Non-Parametric Bayesian Network to estimate monthly maximum river discharge and its potentiality and challenges. The topic is important and would be of great interest to the readers of this field and falls within the scope of HESS. The paper has many grammatical errors and needs lots of editing. I did some of them in the abstract section, but it is not the role of reviewers to edit the full manuscript. The paper is not well organized. This reviewer wants to re-review the article after consideration given to the comments listed below.

We first would like to thank the Reviewer for taking the time to review this work in detail. Below, we addressed the comments received.

1. The authors ought to re-write the abstract so that it briefly presents the problem at hand, objectives of the study, methods used to achieve the objectives in a logical order before presenting a summary of major results and conclusions drawn from the study.

We are a bit surprised by this comment as the information mentioned is already included in the abstract, following the same order mentioned by the reviewer (the lines refer to the first version of the manuscript, version reviewed by the Reviewer):

Knowledge gap: Unclear whether Non-Parametric Bayesian Networks (NPBNs) are suitable tools to predict river discharge, as there are only very few studies using this method in hydrology (Line 2)

Objective: Explore here the potential of NPBNs to reproduce catchment-scale hydrological dynamics. (Lines 2-3)

Methods: 3 different Nonparametric Bayesian Networks (Unsaturated Network (UN-1) and Saturated Network (SN-1) with only hydro-meteorological variables and trained on one catchment; Saturated Network with hydro-meteorological variables and catchment properties (SN-C) and trained on all the catchments. (Lines 4-8)

Following the Reviewer suggestion, we have modified the abstract so that the knowledge gap, objective, and methods are more clearly highlighted in the order suggested. Please, refer to Lines 1-13 of the revised manuscript.

2. The introduction of the manuscript was very poorly written. The reason why you carried out this study does not seem to justify a publication. Try to highlight the regional or national significance of this study, especially since a lot of similar work has been done.

The main objective of this study is to explore and test the potential of Non-Parametric Bayesian Network (NPBN) to reproduce river discharge given its several potential advantages, e.g., the uncertainty quantification is embedded in the model, all the variables can be inferred via conditioning on the remaining variables, knowledge on the relationship between variables can be imposed a priori, information from different catchments can contribute to improve inference, and the computational time is limited. Hence, the significance of the study lies in the appraisal of this specific method rather than in a comparison of regional/national patterns of streamflow. The selection of the study basins, as dictated by the necessity of having a consistent and complete dataset of large number of catchments from diverse climate, then served as actual means to test the method using a large sample of study basins characterized by different environmental conditions. However, the main objective of this paper remains to test NPBNs for their suitability as tools/methods to estimate river discharge. We appreciate the comment of the Reviewer. We have clarify the objective of the manuscript as follows:

Lines 74-79: *Starting from these premises, the main objective of this study is to further explore and test the suitability of NPBNs as a tool to reproduce catchment-scale hydrological dynamics and to explore challenges involved when inferring monthly maximum discharge. More specifically, long-term data from 240 river catchments across the United States from the Catchment Attributes and Meteorology for Large-sample Studies (CAMELS, Newman et al. (2015) and Addor et al. (2017)) data set will be used as actual means to test the utility of NPBNs as descriptive models and to evaluate them as predictive models for monthly maximum river discharge considering the catchments individually and in group, to explore catchment similarity.*

3. The authors should also discuss other algorithms previously used by other researchers to predict the monthly maximum river discharge in the introduction section and explain why only NPBN were chosen for this study?

As discussed in our response to the previous comment, the main objective of this study is to explore the potential of Non-Parametric Bayesian Network to reproduce river discharge. This was an a-priori decision based on the potential advantages of this method. In the introduction, we provided an overview of methods used in hydrology for generation of river discharge values and we divided these methods into process-based models and process-agnostic models.

Highlight the key points of the paper, the innovative part of your work. What differentiates it from other works? Why should the journal publish it?

We thank the Reviewer for this comment.
The key point of the paper is to investigate the applicability of a fully probabilistic process-agnostic approach to predict river discharge generation. Given its several potential advantages, such as the uncertainty quantification embedded in the model (see line 68-73 of revised manuscript for details), we decided to test whether this type of probabilistic model, frequently used in other disciplines for risk and reliability assessment, could be implemented also for generating samples of river discharge. While investigating its suitability for river discharge characterization, we identified some benefits (e.g., embedded uncertainty quantification) and challenges (e.g., Gaussian assumption for bivariate dependence) and we reported them in the Discussion section to incentivize further studies.

In the revised manuscript, we have addressed this comment in the *Introduction*. Please refer to Lines 44-79 in the revised manuscript.

4. The quality of the figures should be improved.

Following the suggestion of the reviewer, we have improved the quality of the figures to increase their readability. At the same time, it was not very clear to us which aspects of the figures need improvement, e.g., resolution, colour codes, legends, content presented.

We modified the bottom panels in Figure 5 to better show the difference in the performances of the networks UN-1 and SN-1. The histograms represent the ratio between the simulated and the observed quantiles for different quantiles value, i.e., 0.05, 0.50, and 0.95. The light grey histograms refer to the network UN-1 and the dark grey histograms refer to the network SN-1

We also modified Figure 6 and now the colorbar of each map to improve readability.

5. Make your conclusion more clear and simplified. Highlight your important result or findings.

We agree with the Reviewer's comment and we have modified the Discussion session to improve it readability. We have divided it in subsections to highlight the key points of discussion. Please refers to the section *"Discussion and Challenges"* , section 6, Lines 430-505

Moreover, in the revised version of the manuscript we have also included the average performance of other river discharge methods found in the literature.

Lines 353-368 – *"In a recent study, Ren et al. (2020) investigated the performances of regression models based on a variety of filter-based feature selection methods to estimate average monthly river discharge in three catchments from the CAMELS data set. The results obtained in terms of NSE ranged from ~0.6 to ~0.8, values similar to average (mean) performance of the network SN-1 (NSE ~0.596) here investigated for maximum river discharge. Kratzert et al. (2019) used CAMELS data set to evaluate the performances of hydrological models. They investigated the performances of Long Short Term Memory (LSTM) network to estimate daily river discharge in 530 catchments and included also, among other models, the performances of the Sacramento Soil Moisture Accounting (SAC-SMA) conceptual model. For the sake of discussion, we look at the performances of LSTM network without catchments attributes and SAC-SMA from Kratzert et al. (2019) for a subset of catchments also analysed in this study. Kratzert et al. (2019) results are available at https://github.com/kratzert/lstm_for_pub. The LSTM network without catchments attributes and the SAC-SMA conceptual model for daily river discharge have an average (mean) performance of NSE ~ 0.603 and 0.598 respectively. SN-1 network, here investigated, for monthly maximum river discharge has an average (mean) performance of NSE ~0.596. In general, NSEs obtained for simulations on a daily temporal scale tend to be lower than the ones on a monthly temporal scale due to the higher amount of observations over a common fixed period of time (Moriasi et al., 2007). However, other studies suggest that for both daily and monthly model simulations a satisfactory performance is given when 0.37<NSE<0.75 (Van Liew et al., 2007). "*

Lines 514-518 – Conclusion: *"The NPBNs individually trained to specific catchments showed potential to reproduce monthly maximum river discharge in a wide range of environments with an average NSE of 0.59 (predictive models), while in the literature the performance of regression models for average monthly river discharge were NSE ~0.6 to ~0.8 (Ren et al., 2020), and the performances for daily river discharge were NSE of ~0.603 and ~0.598 for LSTM network and SAC-SMA model respectively (Kratzert et al., 2019) "*

6. Remove unnecessary "the" from the manuscript.

We thank the reviewer for the suggestion. We have done a thorough grammar check to minimize the amount of placed articles in the revised manuscript.

Specific comments:

We thank the reviewer for the specific comments. Below a summary of the changes made following the reviewer's suggestions.

Abstract:

In line 4 authors wrote UN and SN networks, and then in line 8, they said UN and SN models. Is there any difference between these two? If not, then please use only one for uniformity in the manuscript.P1 L2: "However, few hydrological applications can be found in the literature." This sentence doesn't fit after the prior sentence.

In this study, the Bayesian networks represent the numerical model used to determine the joint probability distribution of the hydro-meteorological variables and catchment characteristics, and then infer from it river discharge. We will replace models with networks to avoid confusion. Thanks for the suggestion.

L2 has been modified as:

Lines 2-3: *"Despite their several advantages, such as the embedded uncertainty quantification and the limited computational time required for the inference process, NPBNs' applications in hydrological studies are still scarce."*

P1 L2: Change "We therefore" to We, therefore," The change has been implemented.

P1 L4: Write the full form of CAMELS first before using its abbreviation. We have implemented the suggested comment.

Lines 5-8: *"Long-term data from 240 river catchments with contrasting climates across the United States from the Catchment Attributes and Meteorology for Large-sample Studies (CAMELS) data set will be used as actual means to test the utility of NPBNs as descriptive models and to evaluate them as predictive models for monthly maximum river discharge."*

P1 L4: Change "one saturated" to "one saturated network" The change has been implemented

P1 L6: What is SN-C? SN-C is the name given to the network that estimates river discharge using also information from the catchments' attributes. We will clarify this in the revised version. We have revised the manuscript as follows:

Lines 11-13: *"Then, we analyse the performance of a saturated network (hereafter SN-C), consisting of the network SN-1 and including physical catchments attributes, to model a group of catchments and infer monthly maximum river discharge in ungauged basins based on the similarity of the attributes.*

P1 L6: Delete "but additionally" The change has implemented. Please, see the response to the comment P1 L6

P1 L8: Change "the attributes similarity" to "the similarity of the attributes" The change has implemented. Please, see the response to the comment P1 L6

P1 L10: Use "," after "analysed" Changed

P1 L14: Use "," after "catchments" Changed

P1 L15: Remove "," before "once" Changed

P1 L15: Remove "," after "discharge" Changed

P1 L16: Change "Despite these advantages, the result also suggest that there are considerable challenges in defining a suitable NPBN, in particular for predictions in ungauged basins." to "Despite these advantages, the result also suggests considerable challenges in defining a suitable NPBN, particularly for predictions in ungauged basins." The change has been implemented as follows:

Lines 22-23 *"[…]the results also suggest considerable challenges in defining a suitable NPBN, particularly for predictions in ungauged basins"*

Please make these changes in the abstract section and revise the whole manuscript for other English grammar and typing errors.

We appreciate the Reviewer's suggestion and will pay extra attention in the revision of the manuscript to avoid grammar and typing errors.

---

## Author Response (AR2)

**Editor**

Please check the reviewers' comments and revise the paper accordingly. Your revision will be sent out for another round of review. Thanks for your contribution.

Dear Editor,

First of all, we would like to thank you for taking the time to review our work in detail. Below the answers to the comments received in blue.

Best,
E. Ragno, also on behalf of M. Hrachowitz and O. Morales-Nápoles

**Report #1 – Referee #2**

We would like to thank one more time the Reviewer for taking the time to review our work in details and providing valuable comments. We are glad the Reviewer is satisfied with our work.

**Report #2 – Referee #3**

As learned from the point-by-point responses and revised manuscript, I would like to recommend a moderate revision before its publication. There are several issues to be further addressed:

We would like to thank the Reviewer for taking the time to review the revised manuscript and the responses in detail. Below, we addressed the comments and suggestions provided. Lines numbering refer to the revised manuscript.

1. The Abstract section is too long. Please rephrase it into less than 250 words.

We thank the Reviewer for the suggestion. We tried our best to shorten the abstract as much as possible. However, we think that shortening it even more will result in removing important information. We managed to reduce it to 330 words (440 in the original version)

Lines(1-20) *Non-Parametric Bayesian Networks (NPBNs) are graphical tools for statistical inference widely used for reliability analysis and risk assessment and present several advantages such as the embedded uncertainty quantification and the limited computational time for the inference process. However, their implementation in hydrological studies is scarce. Hence, to increase our understanding of their applicability and use in hydrology, we explore the potential of NPBNs to reproduce catchment-scale hydrological dynamics. Long-term data from 240 river catchments with contrasting climates across the United States from the Catchment Attributes and Meteorology for Large-sample Studies (CAMELS) dataset will be used as actual means to test the utility of NPBNs as descriptive models and to evaluate them as predictive models for maximum daily river discharge in any given month. We analyse the performance of three different networks, one unsaturated (hereafter UN-1), one saturated (hereafter SN-1), both defined only by hydro-meteorological variables and their bi-variate correlations, and one saturated (hereafter SN-C), consisting of the network SN-1 and including physical catchments attributes. The results indicate that the network UN-1 is suitable for catchments with a positive dependence between precipitation and river discharge, while the network SN-1 can reproduce discharge also in catchments with negative dependence. The latter can reproduce statistical characteristics of discharge (tested via the Kolmogorov-Smirnov*

*statistic) and have a Nash-Sutcliffe Efficiencies (NSE) ≥ 0.5 in ~40% of the catchments analysed receiving precipitation mainly in winter and located in energy-limited regions at low to moderate elevation. The network SN-C, based on similarity of the catchments, can reproduce river discharge statistics in ~10% of the catchments analysed. We show that once a network is defined, it is straightforward to infer discharge and to extend the network itself with additional variables, i.e. going from the network SN-1 to the network SN-C. However, challenges remains in defining a suitable NPBN mainly due to the discrepancies in the time scale of the different physical processes generating discharge, the presence of a "memory" in the system, and the Gaussian-copula assumption used for modelling multivariate dependence.*

2. Introduction: It is not clear that why the catchment-scale hydrologic dynamics, especially monthly maximum river discharge, is selected as the research target. The authors are required to present more literature reviews of the limitation of the above issue. The temporal resolution of monthly is an interesting topic, but how about the research progress in the prediction of monthly maximum river discharge. Besides the proposed NPBN method, can the authors give some examples/limitations of the other approaches for predicting the monthly maximum river discharge? Also, how about the performance of the NPBN method on the daily maximum river discharge prediction? In a word, the research gap of this study needs to be further clarified.

We thank the Reviewer for the comment and we agree on the need to specify that the variable of interest is the maximum daily river discharge in any specific month.
We have modified the manuscript and replaced "monthly maximum" with "daily maximum". To clarify it further, we also modified the title. The new title will be *Applying Non-Parametric Bayesian Network to estimate maximum daily river discharge: potential and challenges.*

The choice of the maximum daily event in a month was based on the fact that we wanted to investigate independent events, i.e., events generated by different weather events (see lines 99-100), and, at the same time, we wanted to keep as many observations as possible from the CAMELS dataset. On the other hand, reproducing daily discharge would have required a completely different approach able to account for the dependence between daily observations (see suggestion on line 476-477 about Dynamic Non-Parametric Bayesian Networks).

We would like also to refer to the discussion included in Section 6 - *Discussion and Challenges* – about the interaction between spatial and temporal scales when trying to model discharge generation using NPBNs. NPBN models the causal relationship between river discharge and hydro-meteorological variables via (conditional) correlation, which, however, is a measure of dependence and does not imply causation. Hence, to implicitly account for the temporal component of the underlying physical processes, we sampled hydro-meteorological variables within a 7-day time window prior to maximum discharge event. However, with this procedure we might have missed some relevant interaction, such as the different response of river discharge to a precipitation event due to soil conditions (see lines 454-469).

Below the main changes made in the manuscripts to clarify the target variables:

Lines 5-8 (Abstract): *Long-term data from 240 river catchments with contrasting climates across the United States from the Catchment Attributes and Meteorology for Large-sample Studies (CAMELS) dataset will be used as actual means to test the utility of NPBNs as*

*descriptive models and to evaluate them as predictive models for maximum daily river discharge in any given month.*

Lines 75-78:*Starting from these premises, the main objective of this study is to further explore and test the suitability of NPBNs as a tool to reproduce catchment-scale hydrological dynamics and to explore challenges involved when inferring maximum daily river discharge in any given month.*

Lines 86-88" *As the objective of this study is to model maximum daily river discharge in any given month from 1980 to 2013, we further process daily hydro-meteorological data as follows: (1) extract the maximum daily discharge for every given months from daily specific discharge;[…]"*

3. Model inputs: Why the previous 7 days is applied in the variable consideration? Why not 14 days or 1 days or others? The authors should perform a sensitivity analysis about the selection of 7 days for the prediction time scale. In addition, the authors assumed that the monthly maximum discharge is mainly driven by monthly maximum precipitation event. It needs more evidences to clarify this assumption. The authors consider several hydro-meteorological data and catchment attributes shown in Table 1 as the model variables. It is strange that soil moisture is not considered in the analysis, and can the authors give necessary explanation?

We thank the Reviewer for the comment. NPBNs model the dependence between the variables via their bi-variate correlation. However, correlation is only a measure of dependence and does not imply causation (Lines 457-458). To ensure the physical link between the observed maximum discharge event of interest and precipitation we looked into precipitation events occurred prior to the discharge event observed (Lines 94-98). In doing so, we assumed that precipitation is the main physical driver for discharge. However, other factors, such as snowmelt, play a role in discharge generation (see Section 2 - *Catchments and data* and *Figure 2* for discussion) and these factors are implicitly included via the bi-variate correlations between discharge and other hydro-meteorological variables, i.e., temperature, solar radiation, water vapour pressure, monthly runoff coefficient, and catchment's attributes (see Section 4 - *NPBN as model for river discharge generation* for details).

The selection of the number of days prior to the maximum discharge event was done based on an a-priori sensitivity analysis (see Lines 94-98). We first looked into the correlation between maximum discharge and both maximum (Supplementary Material Figure S1 top panel) and cumulative (Supplementary Material Figure S1 bottom panel) precipitation over a time windows between 1 and 7 days to understand the catchment behaviour in relation to a single intense precipitation event or a series of less-intense precipitation events. We observed that the correlation is not much affected by the type of precipitation events. Since the maximum precipitation event is included in the cumulative precipitation value, we selected the maximum precipitation event as the variable of interest. Moreover, we observed that the increase in the time window of observation did not affect much the correlation between maximum discharge and maximum precipitation event. This could be justified by the fact that the catchments analysed have a catchment area ≤ 200 km$^2$. However, give the variability in catchment areas between the catchments analysed, we decided to select the maximum precipitation event from a 7-day window, the largest window. We then further investigated whether the maximum precipitation event of the month and the maximum precipitation event observed in the 7-day window prior to the monthly daily maximum discharge event were the same, Supplementary material Figure

2. We observed that this was the case in about 60% of the months except in catchments at high elevation where there is a negative correlation between precipitation and discharge.

We agree with the Reviewer about the importance of soil moisture. As mentioned in Section 4-*NPBN as model for river discharge generation*, we initially included soil moisture as variable. However, NPBNs are data intense and a complete and consistent multivariate datasets are necessary to train and validate the networks. Hence, we selected ESA CCI Soil moisture data because it provides soil moisture since 1978, while SMAP mission was launched in 2016 (https://www.jpl.nasa.gov/missions/soil-moisture-active-passive-smap) and AMSR-E covers only the period from June 2002 to October 2011 (https://disc.gsfc.nasa.gov/datasets/LPRM_AMSRE_A_SOILM3_002/summary). Even though ESA CCI provided soil moisture information over the same time period of the CAMELS dataset, the presence of missing values around the time of the peak river discharge led to multivariate time series too short for training the networks. This is because if in one month one of the variables was missing the entire month was removed from the multivariate time series. Moreover, as mentioned in Section 6-*Discussion and Challenges - Interacting spatial and temporal scales,* the response of soil moisture to climate conditions, e.g., precipitation event or temperature excursion, is quicker compared to runoff generation. Hence, it was difficult to assess the timeframe containing the soil moisture information relevant to the discharge generation process. For this reason we decided to approximate the soil conditions via a monthly runoff coefficient.

We have revised the manuscript and improved the explanation about soil moisture is not among the variables used.

Lines 211-214: *It is worth mentioning that in a preliminary analysis (not shown here) we have tested the use of ESA CCI (https://esa-soilmoisture-cci.org/) remote sensing soil moisture data since measurements are available from 1978, similarly to the CAMELS data set. However, the presence of missing values significantly affected the length of the multivariate data set of hydro-meteorological variables considered for training and testing the networks of interest. Moreover, the coarse spatial resolution and the time lag between the response of river discharge and soil moisture to external input, such as precipitation, led us to rather use here monthly runoff coefficient as proxy for system-state. A more in-depth discussion is presented in section 6 - Discussion and Challenges.*

4. The 240 catchments used in this study are not shown. It is better to provide a sketch map to display the location of 240 catchments. In Fig.1, the site information is confusing and lacks of catchment characteristics.

We thank the Reviewer for the suggestion and we modified Figure 1 accordingly. We based Figure 1 on Newman et al. (2015) and Addor et al. (2017) in which the CAMELS data set is presented, see for example Figure 1 and 2 in Addor et al. (2017). In the revised manuscript, we made US maps bigger so catchments' locations and attributes are more visible. Given the large number of catchments analysed and the limited available space in the main manuscript, specific information, such as catchments coordinates and identification code, are in Supplementary Material Table S1.

References:

*Addor, Nans, et al. "The CAMELS data set: catchment attributes and meteorology for large-sample studies." Hydrology and Earth System Sciences 21.10 (2017): 5293-5313*

*Newman, A. J., et al. "Development of a large-sample watershed-scale hydrometeorological data set for the contiguous USA: data set characteristics and assessment of regional variability in hydrologic model performance." Hydrology and Earth System Sciences 19.1 (2015): 209-223.*

5. Lines 152-158: The review of BNs is suggested to move to the Introduction part.

Thank you for the suggestion, we moved the literature review about BNs to the introduction Lines 56-63.

6. Lines 215-218: I don't think the explanation of soil moisture limitation is convincing. The authors can consider to use SMAP data (https://smap.jpl.nasa.gov/data/) or AMSR-E (https://disc.gsfc.nasa.gov/datasets/LPRM_AMSRE_A_SOILM3_002/summary) for the sources. The missing of soil moisture would have a significant impact on the accuracy of river discharge prediction.

Please see the answer to comment 3.

7. In the model configuration, the traing and validation data sets are not very clear. The posterior distributions of model parameters are also not shown and analyzed. How the model is validated? It needs to give more details.

We thank the Reviewer for the comment. The multivariate distribution function used to derive river discharge via conditioning is defined by a network consisting of nodes (variables of interests) and the connecting arcs (dependence between variables). In the specific case of NPBN, the bivariate dependence between pairs of variables (arcs) is modelled via Gaussian (or Normal) copula. This implies that the multivariate distribution function is a multivariate normal distribution and it is defined by the correlation matrix R, determined via the procedure described in the section 3.1 Lines 176-184. Once the nodes (hydro-meteorological variables and catchments attributes), the arcs, and the parent nodes ordering are determined, the correlation matrix R is uniquely defined. The variables X mentioned in the procedure differs depending on whether the descriptive model or the predictive model is considered. In the case of the descriptive model, the variables X used to determine the correlation matrix R consist of the entire multivariate time series from CAMELS. In the case of predictive model, the variables X used to determine the correlation matrix R consist of the multivariate time series from CAMELS from which 10 years are, randomly, removed (See section 4.1.-*NPBN testing*, Lines 257-258).

After the networks have been constructed and the associated unique correlation matrix R been calculated, we performed 4 different tests. We first tested if the correlation matrix R is a good representation of the empirical multivariate correlation between the hydro-meteorological variables and catchments attributes. Hence, we performed the Cramer-von-Mises test to assess the validity of the Gaussian assumption in modelling the bivariate dependence between pairs. Then, we performed the d-calibration score to assess the quality of the overall multivariate dependence structure. Afterwards, we tested the performance in terms of predictive capacity of the networks via the Nash–Sutcliffe Efficiency coefficient (NSE) for event prediction and the Kolmogorov-Smirnov test for prediction of statistical characteristics. The results are discussed in section 5 *Results* and presented in Figures 5, 6, and 8.

In the manuscript an in-depth discussion about the feasibility of testing all the possible network configurations is included in section 6 *Discussion and Challenges.*

8. The authors refer the literature results of LSTM to prove the superority of the proposed NPBN. As the study region, time period, etc are different from this study. The simple comparison based on the NSE index is not convining. The authors are required to

compare the NPBN method with the previous methods using the same training/validation data. It will make sense.

We thank the Reviewer for the comment. We would like to emphasize the fact that it is out of the scope of this work to show the superiority of the NPBN. We included references to existing studies implementing CAMELS dataset only to put the results we obtained into a broader perspective. Our aim is to increase our understanding of the applicability of NPBNs and extend their use in hydrology. For this reason, a large portion of the study is dedicated to explain the advantages but also the challenges the method present. We do understand the point of view of the Reviewer that in order to prove the superiority of the NPBN a comparison with other methods using the same dataset would be necessary. However, the purpose of this study was to provide a comprehensive analysis of the suitability of NPBNs to derive river discharge via conditional probability given its several advantages in terms of the characteristics of NPBNs.